# Convergent Representations of Computer Programs in Human and Artificial Neural Networks

**Shashank Srikant**[*,1,4]     **Benjamin Lipkin**[*,2]     **Anna A. Ivanova**[1,2,3]
**Evelina Fedorenko**[2,3]     **Una-May O'Reilly**[1,4]
\* Equal contribution
[1]CSAIL, MIT     [2] BCS, MIT
[3] McGovern Institute for Brain Research     MIT-IBM Watson AI Lab
{shash, lipkinb, annaiv, evelina9}@mit.edu, unamay@csail.mit.edu

## Abstract

What aspects of computer programs are represented by the human brain during comprehension? We investigate this question by analyzing brain recordings derived from functional magnetic resonance imaging (fMRI) studies of programmers comprehending Python code. We first evaluate a selection of static and dynamic code properties, such as abstract syntax tree (AST)-related and runtime-related metrics and study how they relate to neural brain signals. Then, to learn whether brain representations encode fine-grained information about computer programs, we train a probe to align brain recordings with representations learned by a suite of ML models trained on code. We find that both the Multiple Demand and Language systems–brain systems which are responsible for very different cognitive tasks, encode specific code properties and uniquely align with machine learned representations of code. These findings suggest at least two distinct neural mechanisms mediating computer program comprehension and evaluation, prompting the design of code model objectives that go beyond static language modeling. We make all the corresponding code, data, and analysis publicly available at https://github.com/ALFA-group/code-representations-ml-brain

## 1   Introduction

Computer code comprehension is a complex task which recruits multiple cognitive processes—from syntactic parsing to mentally simulating programs. Despite the prevalence of this task, the representations of code processed in the human brain during code comprehension remain uninvestigated. Is it possible that common code properties and program semantics are faithfully represented in brain activity patterns when code is read and evaluated? A few prior works have used data derived from functional magnetic resonance imaging (fMRI) and electroencephalography (EEG) to locate physical regions in the brain involved in code-related activities like code comprehension and debugging [Siegmund et al., 2017, Floyd et al., 2017, Peitek et al., 2018, Castelhano et al., 2019, Ivanova et al., 2020, Liu et al., 2020, Ikutani et al., 2021, Peitek et al., 2021], code writing [Krueger et al., 2020, Karas et al., 2021], and data structure manipulation [Huang et al., 2019]. These studies have helped determine whether code-related activities join other activities supported by those brain regions, such as working memory (processed by a set of brain regions known as the Multiple Demand system) or language processing (processed by the Language system–another set of brain regions). While these results improve our understanding of the brain regions involved in code comprehension, it still remains unclear what specific code-related information these regions encode. For example, does the response in a region seen during code comprehension encode specific syntactic or semantic code properties? Do responses from multiple brain regions correspond to the same set of properties? Or, are different code properties encoded in different regions?

36th Conference on Neural Information Processing Systems (NeurIPS 2022).

**A possible approach and its limitation.** One way to learn what information is encoded in the brain is to decode a code property of interest from recordings of brain signals when reading code (through fMRI or EEG). Being able to decode the property accurately from a specific region of the brain establishes that information related to that code property is faithfully represented in that brain region. A question central to such a decoding analysis is the choice of the target code property–what code properties should be investigated? We can hand-select a set of fundamental properties of code and test if they can be decoded. While helpful, such a set will not preclude other, more complex aspects of code possibly being encoded.

**ML models of code as a tool to reverse engineer what is encoded.** To address the limited scope of hand-selected properties, we look to machine learning (ML) models trained on code. Dubbed *code models*, they are trained on large corpora of code, in an unsupervised manner, to learn *ML model representations* of computer programs. Code models are increasingly being used in software engineering workflows [Allamanis et al., 2018a], and have been shown to perform well on tasks like code summarization [Alon et al., 2019], detecting variable misuse [Bichsel et al., 2016], and more recently, code auto-completion [Chen et al., 2021]. These representations have been shown to encode and describe complex code properties [Bichsel et al., 2016, Allamanis et al., 2018b, Srikant et al., 2021]. Successfully decoding these code model representations from brain activity data then allows us to probe whether complex code properties are also encoded in the brain.

Besides serving as a tool to reverse engineer what is encoded in different brain regions, another distinct advantage of decoding code model representations is its potential to serve as a tool to reverse engineer our cognitive processes. It is currently unclear what mechanisms drive code comprehension. If we find one class of ML models (say, masked language models) to be more predictive than another (say, autoencoders), it is reasonable to suspect that our brains optimize objectives more similar to that of masked-LMs than that of autoencoders when comprehending code. As a corollary, a poor correspondence between the information encoded by our brains and ML models suggests the possibility of unexplored neural architectures and objectives which may better model our cognition, which in turn may outperform extant ML models. Yamins et al. [2013] first showed how information encoded in our visual system resembles what convolutional neural networks learn when trained to recognize images. We attempt to establish a similar correspondence between code models and the human brain in comprehending code.

**Why code comprehension?** We focus on code comprehension in this work because very little of this important skill has been analyzed from a cognitive neuroscience perspective while steady advances are being made in training ML models to understand code and increase programmer productivity [Fedorenko et al., 2019, Hellendoorn and Sawant, 2021]. Extant ML models for understanding programming are direct adoptions of the state-of-the-art in language processing research. However, recent works in neuroimaging like Liu et al. [2020] and Ivanova et al. [2020] suggest that code comprehension does not share the same neural bases as natural language comprehension. Do code models then mimic human cognition of programs? If not, can we consider other model architectures and training objectives which are directly inspired by results from neuroscience? Our work provides early considerations towards these directions.

Further, characterizing the representations of different code properties in the human brain can inform us about the nature of human algorithmic problem solving more generally. We can possibly characterize signatures of brain activity corresponding to fundamental logic operators like iterative reasoning, conditional reasoning, retrieving calculations that were previously computed, *etc.* seen in individuals when solving carefully designed code comprehension tasks. Such signatures will then allow us to characterize other tasks which involve similar logic operators, but which are not naturally described as computer programs *e.g.* general logic reasoning, problem solving, and more generally, our ability to employ abstractions and form concepts [Rule et al., 2020].

**Our setup.** We introduce a framework to evaluate the code properties encoded in human brain representations of code by analyzing fMRI recordings of programmers comprehending Python programs. We present two means of proceeding— one, probing of brain region representations for specific code properties, and two, analyzing the mapping of these representations onto various code models with differing model complexity. We utilize the publicly available dataset from Ivanova et al. [2020] for all our analyses as it offers high quality, granular brain response data on code comprehension stimuli controlled across multiple code properties. We train affine models on brain representations to predict hand-selected code properties which express syntactic and semantic behavior of programs. Similarly,

we predict code representations obtained from a suite of code models. We investigate the effects that brain regions, the nature of code properties, and the complexity of the code models have on the accuracies of these prediction tasks.

**Key findings.** We explore two questions: (1) Do brain systems encode specific code properties? Are there differences in how well each brain system encodes each code property? (2) Do brain systems encode more complex properties of code, derived from the representations of code models? To answer these, we perform three critical comparisons: between code properties and code models, code models and brain systems, and brain systems and code models. We further provide a preliminary analysis into the relationship between code models and the brain in the context of what properties code model representations encode. We find the Multiple Demand system and the Language system consistently encode both hand-selected code properties and data-derived code model representations, using features which cannot be explained by only low-level visual characteristics of the code. Within this set, the Multiple Demand system most effectively decodes runtime properties of code like the number of steps involved in a code's execution, while the Language system decodes syntax-related properties like the number of tokens in a program and the control structures present in it. These results improve our understanding of the functional organization of the two brain systems—MD and the Language systems—that we study, and provide initial evidence for incorporating the roles of these two distinct brain systems in the design of training objectives of code models.

We provide an open-source framework to replicate our experiments, and we release our data and analysis publicly. Link - `https://github.com/ALFA-group/code-representations-ml-brain`. This should enable authors from other neuroimaging studies or code model developers to collaborate and analyze data across these works, which will also help amortize the high costs of carrying out such experiments.

## 2   Related Work

Of the prior works that have investigated the neural bases of programming through fMRI and EEG techniques [Siegmund et al., 2017, Floyd et al., 2017, Peitek et al., 2018, Castelhano et al., 2019, Huang et al., 2019, Krueger et al., 2020, Ivanova et al., 2020, Liu et al., 2020, Ikutani et al., 2021, Peitek et al., 2021] and through behavioral studies [Prat et al., 2020, Casalnuovo et al., 2020, Crichton et al., 2021], the following probe brain recordings for program properties encoded in them.

Floyd et al. [2017] learn a linear model to successfully classify whether an observed brain activity corresponds to reading code or reading text. Ikutani et al. [2021] study expert programmers and show that it is possible to classify code into the four problem categories–math, search, sort, and string from the brain activations corresponding to the code. Similarly, Liu et al. [2020] classify whether a brain signal corresponds to code implementing an `if` condition or not. Peitek et al. [2021] analyze correlations between brain recordings of participants reading code and a set of code complexity metrics.

In testing for code properties, our work uses a similar methodology (a linear model trained on fMRI data), but we evaluate a larger set of static and dynamic code properties, often reflecting key programming constructs like control flow. Further, we perform these tests in the brain regions identified by Liu et al. [2020] and Ivanova et al. [2020] as being responsive specifically to code comprehension, offering finer insight into the content of these specific regions' representations. In addition, we study representations generated by a suite of ML models with varying complexity and compare those learned representations to brain representations.

Brain representations have also been studied in domains like natural language, vision, and motor control. Among related works in natural language, a domain that resembles programming languages, Mitchell et al. [2008], Pallier et al. [2011], Brennan and Pylkkänen [2017], Jain and Huth [2018], Gauthier and Levy [2019], Schwartz et al. [2019], Wang et al. [2020], Schrimpf et al. [2021], Cao et al. [2021], Caucheteux et al. [2021], Toneva and Wehbe [2019] have studied brain representations of words and sentences by relating them to representations produced by language models. While the broader tools we use to investigate these representations, like multi-voxel pattern analysis (MVPA), are similar to some of these prior works, our focus is on properties specific to code and not natural language.

# 3 Background

We provide a brief background on fMRI signals as a proxy for brain representations and describe the brain systems that we probe in this work.

**Measuring brain activity with fMRI.** Functional magnetic resonance imaging (fMRI) is a brain imaging technique used to measure brain activity in specific brain regions. When a brain region is active, blood flows into the region to aid its processing. An MRI machine measures this change in blood flow, and reports BOLD (blood oxygen level dependent) values sampled at the machine's frequency [Glover, 2011, usually 2 seconds]. The smallest unit of brain tissue for which BOLD signal is recorded is called a voxel (an equivalent of a 3D pixel); it comprises several cubic millimeters of brain tissue. For our analyses, we select subsets of voxels belonging to specific brain systems. Following common practice, the parameters of a general linear model, fit to time-varying BOLD values, are used as a measure of the overall activation in each voxel in response to a given input. It is the values of these parameters that, in concordance with common practices in the neuroscience community, can be accessed as brain representations.

**Brain systems.** A system of brain regions can span different areas of the brain but behaves as a holistic unit, showing similar patterns of engagement across a given cognitive task. We probe the following systems in our work: (a) **Multiple Demand (MD) system:** this system of regions is known to engage in cognitively demanding, domain agnostic tasks like problem solving, logic, and spatial memory tasks. Liu et al. [2020] and Ivanova et al. [2020] reported that this system is active during code comprehension. (b) **Language system (LS):** this system responds during language production and comprehension across modalities (speech, text) and languages (across 11 language families, including American sign language). (c) **Visual system (VS):** these regions respond primarily to visual inputs. (d) **Auditory system:** these regions respond primarily to auditory inputs.

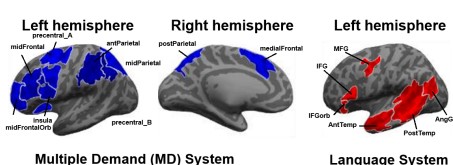

Figure 1: The approximate locations of MD and the Language systems in the human brain. The regions depicted are used as a starting point to functionally localize these systems in individual participants.

We probe the Visual system and Auditory system since they are involved in general perception. While we do not expect activity in Auditory system, we expect the Visual system to reflect low-level visual properties of the code (*e.g.* code length and indentation to reflect code-related properties). See Appendix A for a detailed description of these brain systems along with relevant references.

# 4 Brain and Model Representations

We describe in this section the method we follow to gather representations of code in the brain (Section 4.1), evaluate the different code properties they encode (Section 4.2), and how we compare brain representations to those generated by ML models (Section 4.3).

## 4.1 Brain representations and decoding

We provide a summary of how we process activation signals in the brain elicited by code comprehension, to probe whether they encode any specific code properties. We provide details in Appendix B.

**Dataset.** We use the publicly available brain recordings released as part of the study by Ivanova et al. [2020] (MIT license). It contains brain recordings of 24 participants, each of whom gave consent and is not personally identifiable according to Ivanova et al. [2020] protocol requirements, reading 72 programs from a set of 108 unique Python programs. The 72 programs were presented in 12 blocks of 6 programs each. These programs were 3-10 lines in length and contained simple Python constructs, such as lists, *for* loops and *if* statements. A whole program was presented at once, and the task required participants to read the code and mentally compute the expected output, press a button when done, and select one of four choices presented to them which matched their calculated output.

**From dataset to brain representations.** The original dataset contains 3D images of the brain of each participant. Each voxel value in these images is an estimate of the response strength in this voxel when a particular code (or sentence) problem is presented. To determine which brain systems contain

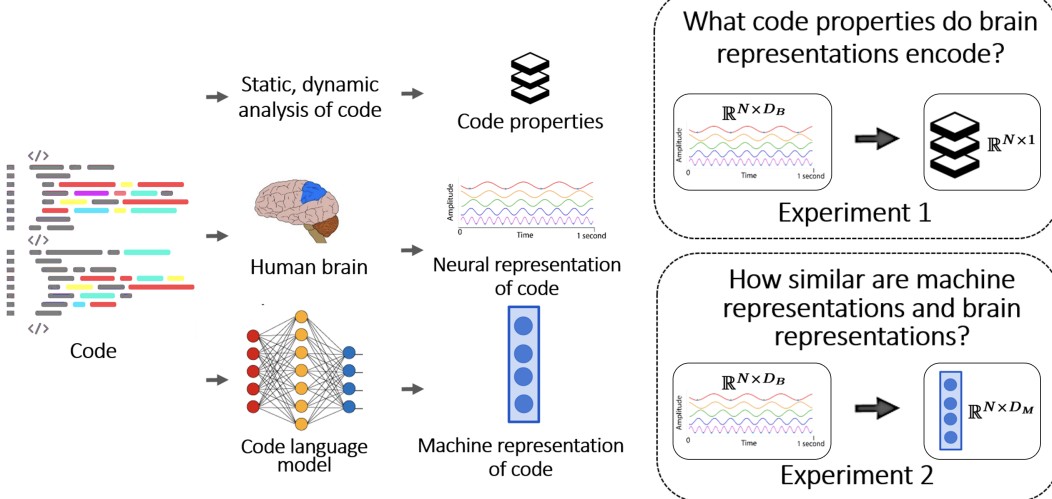

Figure 2: **Overview.** The goal of this work is to relate brain representations of code to (1) specific code properties and (2) representations of code produced by language models trained on code. In Experiment 1, we predict the different static and dynamic analysis metrics from the brain MRI recordings (each of dimension $D_B$) of 24 human subjects reading 72 unique Python programs ($N$) by training separate linear models for each subject and metric. In Experiment 2, we learn affine maps from brain representations to the corresponding representations generated by code language models (each of dimension $D_M$) on these 72 programs.

information about particular code properties, we focus our analyses to four systems – MD, Language, Visual, and Auditory (Section 3). A vector of voxels' activation values in each brain system is then taken to constitute that system's *representation* of a computer program and serves as an input to all our analyses. See Appendix B for details on data processing and voxel selection.

**Analyzing brain representations of code.** We probe brain representations from each participant separately. We do not average data across participants since the regions which respond to any task (comprehending code in our case) need not align anatomically. For each of two experiments—decoding different code properties, and mapping to code model representations—we train ridge regression/classification models which take as input normalized brain representations per participant. We hence learn 24 different regression models, for each code property or code model (one per participant), and then report the mean performance of these models across participants. This procedure is also referred to as multi-voxel pattern analysis (MVPA) [Norman et al., 2006]. Linear models are conventionally preferred for probes into brain representations since there has been evidence supporting the idea that other brain areas linearly map information from such brain representations [Kamitani and Tong, 2005, Kriegeskorte, 2011]. We choose a linear model primarily to control for over-fitting in light of the relatively small dataset.

**A remark on data scarcity.** For each participant, we train a linear regression/classification model with L2-regularization for on unique cross-validated leave-one-run-out folds of the 72 programs they attempted (or 48 programs when sentences are removed). On the order of 1000 voxels were selected from each brain region responding to any given program per participant, thus resulting in a feature set of dimensions $72 \times 1000$. Such dimensions allow for strong statistical tests to support the robustness of the predictions made. As a baseline for the model predictions, we use the accuracy of a null permutation distribution generated from sampling 1000 random assignments of the labels.

## 4.2 Code properties

We attempt to decode the following code properties from brain representations. See Appendix B for details. (a) **Code vs. sentences:** classify whether an input stimulus is a code or an English sentences describing a code (referred to as *sentences*). See Figure B, Appendix B for an example. (b) **Variable language:** classify whether a program contains variable names written in English or Japanese (written in English characters). (c) **Control flow:** predict whether a program contains a loop (`for` loop), a branch (`if` condition), or has sequential instructions. (d) **Data type:** predict whether a program contains string or numeric operations. (e) **Static analysis:** predict static properties of a program like *token count*: number of tokens in the program, *node count*: number of AST nodes, *cyclomatic complexity*, and *Halstead difficulty*. The latter two metrics have been used by software engineering practitioners to quantify the complexity of code, and to quantify the difficulty a human

would experience when comprehending code respectively. We defer predicting other advanced static analysis metrics such as tracking abstract interpretation joins, data flow analysis-related metrics, *etc.* to future work. (f) **Dynamic analysis:** predict information about a code's execution behavior like *runtime steps*: number of instructions executed in the program, and *bytecode ops*: number of bytecode operations executed in running the program.

**Program length as a potential confound.** Since the properties we examine can also potentially be differentiated using program length and other low-level code features, it is a potential confound in our experiments. We measured the inter-correlations of these properties, and their correlation to the number of tokens in the program (program length; see Appendix H). We expectedly found the four *static analysis* properties to be highly correlated to each other and to *bytecode ops*. We hence use one representative metric each from the two categories of properties for the rest of our analysis–*token count* for *static analysis*, and *runtime steps* for *dynamic analysis*. Importantly, the other properties we examine cannot be explained by program length alone, and therefore program length is not a confound in our experiments.

**Mapping to code properties.** The brain representations (Section 4.1) are mapped to each of the code properties by training a ridge regression (for *static analysis* and *dynamic analysis* properties; continuous values) or a classification model each for every participant-property pair. To evaluate model performance, we use classification accuracy when the predicted values are categorical (*e.g.* string vs. numeric data types), and the Pearson correlation coefficient when the predicted values are continuous (*e.g.* number of runtime steps). We choose Pearson correlation over RMSE, the canonical distance metric for continuous values, for its simplicity and interpretability. See Appendix J for the RMSE results, which lead to the same conclusions as the correlation results. When testing for the significance of these predictions, we perform false discovery rate (FDR) correction for the number of brain systems tested and the number of properties tested. See Appendix B for a detailed description of model hyper-parameters, cross-validation settings, and significance testing.

### 4.3   Model representations and decoding.

We evaluate a bench of unsupervised language models, spanning from count-based language models to transformer neural networks [Vaswani et al., 2017]. These models were all trained on large (∼1M programs) Python datasets [Husain et al., 2019, Puri et al., 2021]. We use the output of the trained encoders (raw logits) in each of the neural network models as representations of the code input to the model. We vary the general complexity of these models to test whether that variation is meaningful in establishing the quality of brain to model fits. Model complexity here is the number of a model's learnable parameters. We evaluate the following models, ordered by their increasing model complexity: simple frequency-based language models—*bag-of-words*, *TF-IDF*; auto-encoder based unsupervised models—*seq2seq* [Sutskever et al., 2014], *CodeTransformer* [Zügner et al., 2021], *CodeBERT* [Feng et al., 2020], *CodeBERTa* [HuggingFace, 2020]; auto-regressive models with similar model complexity—*XLNet* [Yang et al., 2019], *CodeGPT* [Microsoft, 2021].

**Baseline: Token projection model.** We compare the results of the above models against an aggressive baseline (relative to the null-distribution labeling baseline), a *token projection model* provided by using unique Gaussian-distributed random vectors for the token embeddings in a vocabulary, and returning the sum of these token embeddings across a program. The resultant embedding is not transformed by any model or any weights–it instead serves as a proxy for the tokens that appear in the program. The results of this baseline model should be interpreted as the level of performance achievable from the presence of tokens alone with no semantic or syntactic information.

**Mapping to code model representations.** The brain representations (Section 4.1) are mapped to code representations by training another set of ridge regression models to learn an affine map, and a ranked accuracy metric is used to compare outputs. Ranked accuracy scores are commonly used in information retrieval where several elements in a range are similar to the correct one. In our case, the top-ranked prediction by the linear model indicates the closest fit (Euclidean distance) to the code model's representation. When reporting result significance, we perform false discovery rate (FDR) correction for the number of brain systems and the number of models. See Appendix B for a detailed discussion of the implemented code models and the corresponding metrics. All experiments in this work were run on a single 8-core laptop in under an hour following setup.

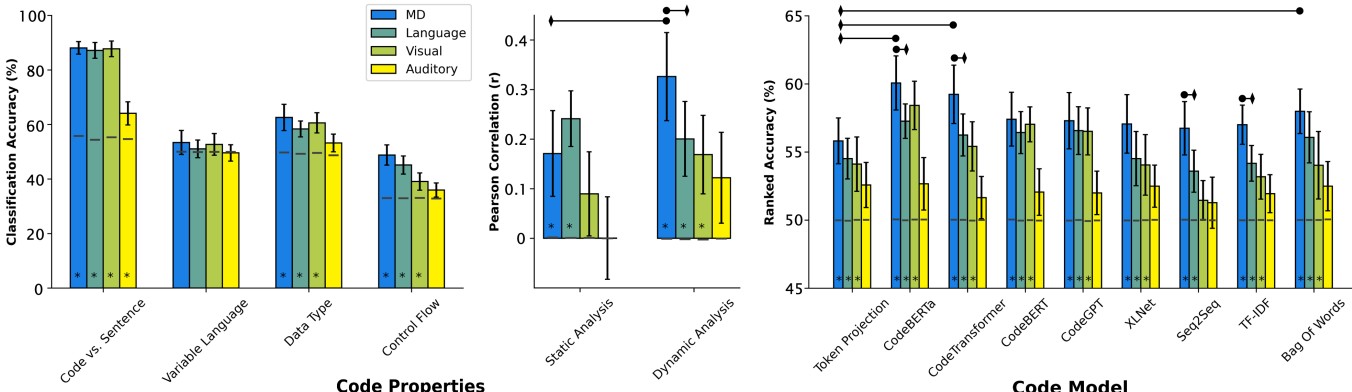

Figure 3: Affine models are learned on brain representations to predict each of the code properties described in Section 4.2, and a collection of code models described in 4.3, for each of the 24 participants. The mean decoding score across subjects is shown here, and error bars reflect the 95% confidence interval of the mean subject score. A solid line on each bar presents the empirical baseline for a null permutation distribution on shuffled labels. All decoding scores were compared to this permuted null distribution using a one-sample $z$-test, and the significance threshold was defined at $p < 0.001$; false-discovery-rate-corrected for the number of tests in each panel (FDR). Statistically significant results are denoted with a $*$, marked at the base of the bars. Additionally, ●-capped lines denote selected significant paired $t$-tests ($p < 0.05$; FDR).

## 5 Experiments & Results

Our experiments address two research questions:

- **Experiment 1.** How well do the different brain systems encode specific code properties? Do they encode the same properties?
- **Experiment 2.** Do brain systems encode additional code properties represented by computational language models of code?

### 5.1 Experiment 1 - How well do the different brain systems encode specific code properties? Do they encode the same properties?

We analyze the classification models and regressions trained on brain representations to predict each of the code properties described in Section 4.2. The results of our analyses are summarized in Figure 3. The classification and regression tests are marked on the x-axis of the left and right subplots respectively; the classification accuracy or Pearson correlation for each of the tasks is marked on the y-axes. We plot dynamic and static properties separately from the others because their baselines are different due to a difference in the similarity metric (classification accuracy vs. Pearson correlation). The baselines for the categorical code properties differ from each other due to variation in the number of target classes.

**Auditory and Visual systems.** The Auditory system and the Visual system serve as negative and baseline controls for the other systems. Since our code comprehension task is visual, we do not decode any meaningful information between programs from Auditory system, although we do observe decoding of *code vs sentences*, perhaps explained by previous work showing auditory cortex activation during silent sentence reading [Perrone-Bertolotti et al., 2012]. The Visual system serves as a baseline for low-level visual features of the code (the layout and indentation of the code, the presence of letters and alphabets in the programs, *etc.*). In Analysis 2, we show that the MD and the Language systems encode more than such visual features. The MD and the Language systems yield the following observations.

**Analysis 1 - How accurately are different properties predicted by MD and LS?** The MD and LS together decode all the properties well above chance barring variable language. The variable language finding is consistent with Ivanova et al. [2020], who show a lack of any significant difference in the aggregate neural activity between English and Japanese variables names—variable names seem to not be encoded any differently in the context of programs we study. To analyze the other results, we use paired two-sample $t$-tests ($p < 0.05$; FDR-corrected) and examine whether for a

given property, any one brain system decodes it significantly more effectively than another (details in Table 8, Appendix E). We find that the MD system decodes the *dynamic analysis* property better than the Language system. We additionally test if any brain system has a preference for a specific code property over another. We find that the MD decodes the *dynamic analysis* property significantly better than the *static analysis* property. These findings establish the role of the MD system in encoding code-simulation and execution related information—an important of aspect of code comprehension.

**Analysis 2 - Multi-system partial regression analysis.** The decoding performance of the VS is comparable to that of the MD and LS (Figure 3). To assess the possibility that all three systems - MD, LS, and VS encode the same properties (all potentially related to low level program features), we employ a multi-system partial regression analysis. For each brain system, MD and LS ($S_i$), we train two models–one which decodes from VS, and another which decodes from $S_i$+VS. If the difference in the prediction accuracies between the two models is significant, we conclude that $S_i$ encodes at least some information which is orthogonal to the information encoded by the Visual system. This method is similar to a variance-partitioning analysis which is often employed in encoding models, *e.g.* Deniz et al. [2019]. For all core properties, *control flow*, *data type*, *dynamic analysis*, and *static analysis*, we find the MD to encode information orthogonal to the VS. For *control flow* and *static analysis*, the LS also encodes information orthogonal to the VS. This suggests that low-level code properties are insufficient to explain the key results from Experiment 1. Other combinations in the regression model reveal that the MD encodes information orthogonal to the LS when predicting *code vs. sentences* and *dynamic analysis*. Detailed results are tabulated in Table 15, Appendix F.

**Key learnings from Experiment 5.1.** All core code properties are decodable from the representations of high-level brain systems, and this information is beyond that which can be explained from low-level visual information alone. Although no property is exclusively encoded in any one brain system, the MD system significantly encodes *dynamic analysis*-related properties–more than what the LS encodes, and more than *static analysis* properties. Similarly, we find evidence for the LS to also significantly encode *static analysis* and *control flow* related properties. These are new results on the nature of code properties different brain systems seem to process and encode. To explore properties which may not be specified by the set we investigate in this experiment, in the following section, we leverage code models as hypothesis-free proxy representations for code syntax and semantics, and see if any one system preferentially encodes a code model.

## 5.2 Experiment 2 - Do brain systems encode additional code properties encoded by computational language models of code?

We train ridge regression models with brain representations of programs from specific brain systems to predict code model representations of the same programs. The set of results from this experiment are summarized in Figure 3.

**Auditory and Visual systems.** Similar to Experiment 1, these systems perform as expected, with the Auditory system exhibiting the lowest decoding performance across code models, and the Visual system acting as a proxy for low-level information (Table 10, Appendix E).

**Analysis 1 - How well do brain representations in MD and LS map to code model representations?** We find that the MD and LS map to all the models in our suite significantly more accurately than the null permutation baseline (Figure 3). Further, we find that the MD ranked accuracy is higher than LS for *CodeBERTa*, *CodeTransformer*, *seq2seq* and *TF-IDF* (differences evaluated using two-sample $t$-tests; $p < 0.05$; FDR-corrected. See Table 10, Appendix E).

**Analysis 2 - The effect of model complexity on decoding to code models.** We investigate the impact model complexity has on the performance of the mapping between brain and code representations.

We compare each of these code models against the *Token Projection* baseline model, which only encodes the presence of specific tokens, with no contextual or distributional information. We find that the MD system maps to all the models more accurately than the *Token Projection*, but this is not observed for the LS. In a set of paired two-tailed $t$-tests ($p < 0.05$; FDR-corrected), we find that the MD maps to three models: *CodeBERTa*, *CodeTransformer*, and *bag-of-words* significantly more accurately than to the *Token Projection* model (Table 11, Appendix E). Curiously, since all but 3 of these mappings do not significantly surpass the model which can be explained using only token-level information, these data suggest that the brain signals we access primarily encode token-level

information. To investigate this further, we analyze the correspondence between brain representations and code models in the context of the code properties they encode.

**Analysis 3 - Code model and brain representations in the context of code properties.** We first evaluate whether the different code properties we investigate in this work can be decoded from code model representations. We find that all the code properties are strongly encoded in all models (Table 4, Appendix C.2). Since we have computed the mapping accuracies from code models → code properties (mentioned above), and from brain representations → code properties (Tables 1 & 2, Appendix C.1), we compute how well brain representation decoding results map to code model decoding results using Spearman rank correlation (details in Appendix G). We find two clusters in decoding performance across the code properties, each reflecting a distinct computational motif (Table 17, Appendix G). We see perfect rank correspondence between the $z$-scores of the decoding results for the MD and three transformer architectures: *CodeBERTa*, *CodeBERT*, and *CodeTransformer*, and between LS and two token-based models: *bag-of-words* and *Token Projection*.

**Key learnings from Experiment 5.2.** The MD decodes representations from four models of code significantly more accurately than LS, providing evidence that some aspects of code are more faithfully represented in the MD. A follow up partial regression analysis, as in 5.1, reveals that for most models, MD encodes information orthogonal to the LS, and each system encodes information above low-level VS (Table 16, Appendix F).

Analyzing the correspondence between code models and brain representations based on the code properties they each encode reveals that the MD system maps preferentially to complex code models, encoding more than just token-level information. We discuss these results further in the following section.

# 6 Discussion

Through this study, we learn what computer program-related information can be decoded from the brain, and which brain systems primarily encode that information.

**Brain representations and code properties.** We show that the MD system preferentially encodes *dynamic analysis*-related properties when compared to other brain systems and other properties. Further, we find the Language system encodes syntax-related properties like *control flow* and *static analysis*. These findings are complementary to the results from Liu et al. [2020] and Ivanova et al. [2020]–they show how the MD system is recruited in both mentally simulating code and comprehending code. They however do not find any consistent response in the Language system to either code simulation or comprehension. Our results show that despite not exhibiting significant average responses, these systems do encode code-specific properties, improving our understanding of these brain systems' functional organization.

**Brain representations and code model representations.** Another key contribution of our work, from Experiment 2, is demonstrating that it is possible to map brain representations to representations learned by code models. In particular, we observe encoding of the properties represented by code models in the MD and LS, with 4 models more accurately mapped from MD. This is particularly noteworthy since these models, which are trained on source code symbols, can be mapped more faithfully from the representations of a network implicated in problem-solving than one associated with composition in languages.

We also considered model complexity as a relevant feature, and note that the MD and LS map to a combination of token projection embeddings almost as well as to complex models like *XLNet*. One plausible explanation for this surprising result is that the program stimuli are simple enough to allow the different properties evaluated in our work (*control flow*, *data type*, *etc.*) to be discerned from token level information alone (as validated in Experiment 1), which is likely why the *Token Projection* model also predicts these properties very well (Table 4, Appendix C.2). Taken together, these data suggest that the information being decoded from brain activations in these two regions is driven at least by the information conveyed by tokens in the programs. This finding is notably consistent with work in the field of natural language which show swaths of cortex are predicted primarily by the token-level properties of sentence stimuli [Toneva et al., 2022].

While we see that information encoded in both the MD and the Language systems are driven by token-level information, a clearer trend emerges when evaluating brain-code model mapping in the

context of specific code properties. We find that the MD shows decoding performance consistent with complex model architectures, suggesting that it may encode more than just token-related information, as complex code models capture contextual information as well. This is a new result, which along with Experiment 1, suggest distinct roles for the LS (purely token-level) and MD (more complex interactions) with respect to computer code comprehension and execution.

**Way forward.** Our findings have the potential to improve our understanding of the organization of the human brain, which can in turn lead to the design of better code models. In computer vision, results by Tschopp et al. [2018], Schrimpf et al. [2020a] show how deep network architectures that mimic the visual system exhibit superior image classification rates on image recognition tasks. Our findings prompt yet another reconsideration of the current design of ML models of code. Extant code model architectures do not explicitly model the Multiple Demand system in any way–they only model syntactic information and infer dependency information from program ASTs. Taking inspiration from the role of the Multiple Demand system we identified in this work, modeling dynamic runtime information as well as static code structure should be explored. See Srikant and O'Reilly [2021] for a related discussion.

We also provide initial results supporting the ability to decode specific and fundamental code-related primitives like control flow information. This sets us up to study other complex human cognitive processes which involve such primitives, but which are not naturally described in terms of code, like general problem solving, employing hierarchical and complex decision making strategies, *etc*. Such a study also promises to support the recent proposal of a child as a hacker Rule et al. [2020].

Our work also promises to enhance *code prosthetics*–artificial interfaces that can help the physically challenged engage with programming environments. Such systems generally rely on designing and constructing brain decoders–models that convert brain activity to electrical impulses modulating external devices, which remains an open challenge. See the discussion in Nuyujukian et al. [2018] and Andersen et al. [2019] for details.

**Limitations.** The average program in any software project exhibits non-trivial control and data dependencies, object manipulation, function calls, types, and state changes. However, the programming tasks in Ivanova et al. [2020] are short snippets of procedural code with limited program properties. Responses to longer programs, with more complex properties and across multiple languages, should be studied on a larger number of participants in the future, in order to build on the trends provided by our work. Further, while there are multiple equally important aspects to programming like designing solutions, selecting appropriate data structures, and writing programs, here we have chosen to study only a single specific activity—code comprehension. Future work should explore these other aspects. Our results do not allow us to infer whether the MD and LS are driven by the same underlying features of code that are used to discriminate between code properties and code models, so future work might consider an encoding analysis. Finally, as with all neural decoding analyses, extracting information from the mental states of participants should be done with caution to ensure it is not used for any exploitative purpose.

**Acknowledgement.** This research was partially supported by NSF grant 1744809. SS and UMO'R were supported by a grant from the MIT-IBM Watson AI Lab. AI was supported by a postdoctoral fellowship from the Quest Initiative. EF was additionally supported by NIH grants DC016607, DC016950 and NS121471, the Quest Initiative, and research funds from the McGovern Institute for Brain Research, the Brain and Cognitive Sciences department, and the Simons Center for the Social Brain. We thank our anonymous reviewers for their thorough and helpful feedback.

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
