# A  Brain regions

**Regions of Interest (ROIs).** We investigate the representations of code in well-studied systems of brain regions–the MD system, Language system, Visual system, and the Auditory system (details below). A *region* (also referred to as *parcel*) here denotes a contiguous chunk of brain mass involved in a cognitive task. A *system* of regions (also referred to as a *network*) can comprise multiple disjoint regions that exhibit shared activity patterns across a range of tasks. For many cognitive tasks, a region marks only the approximate location of voxel populations involved in a cognitive task. *Functional* ROIs (fROIs) are voxels within these broad regions that respond most strongly to a given task (language, working memory, *etc.*). The use of fROIs enables accounting for the exact anatomical locations of these task-sensitive voxels, which vary across individuals.

**Multiple Demand (MD) system.** Located in the prefrontal and parietal areas of the brain, this system of regions is active in a host of tasks requiring working memory and general problem solving skills, including math and logic [Duncan, 2010, Fedorenko et al., 2013, Amalric and Dehaene, 2019].

**Language system.** These regions have been identified to respond to both comprehension and production of language across modalities (written, speech, sign language), respond to typologically diverse languages (45 languages, from across 11 language families), form a functionally integrated system, reliably and robustly track linguistic stimuli, and have been shown to be causally important for language [Clark and Cummings, 2003, Fedorenko et al., 2010, Hu et al., 2021, Blank and Fedorenko, 2017, Newman et al., 2015, Ayyash et al., 2021].

**Visual system.** These regions in the occipital lobe of the brain respond to visual input, ranging from low-level features like lines and edges, through intermediate categories like shapes, letters, and numbers, through higher order structures like faces and scenes. [Hubel and Wiesel, 1959, Polk et al., 2002, Epstein and Kanwisher, 1998, Kanwisher et al., 1997].

**Auditory system.** The auditory system is located in the superior temporal region of the brain. This region uniquely encodes pitch, speech, and music, but is not involved in high-level language comprehension and production [Norman-Haignere et al., 2015, 2019]. In our experiments pertaining to programming language comprehension, we use the activity seen in the auditory system as a negative control.

# B  Method - Details

**Selecting brain representations.** For each trial in the fMRI experiment, stimulus responses in each voxel were extracted from the parameters of a General Linear Model (GLM) fit to the time-varying BOLD signal in which each experimental condition was modeled with a boxcar function convolved with the canonical hemodynamic response function (HRF). For the localizer experiments, conditions were modeled as the entire block. For the Python program comprehension experiment, individual programs were modeled using the period from the onset of the code/sentence problem until the button press. The predictors for the GLM included trial ID (equivalent to problem ID), run number, and motion regressors. The voxels were then filtered using gray-matter masking and (for MD and the Language systems) network localization. Although fMRI measurements return whole brain responses, only a thin layer of cortex dubbed gray matter contains BOLD signal of interest to these analyses. Gray matter voxels were selected using a Bayesian segmentation of the anatomical brain image into standard tissue types, and then returning the set of indices where the posterior gray matter probability exceeds 0.70 [Ashburner et al., 1999]. Next, these sets of voxels were filtered separately for each of the brain regions outlined in Appendix A. For the visual and auditory networks, primary sensory areas were identified using an anatomical atlas [Rolls et al., 2020]. For the MD and the Language systems, voxels were functionally localized as those containing the top 10% of responses to their respective functional localizer tasks, as described in Ivanova et al. [2020]. See Fedorenko et al. [2010] for a discussion of the functional localization approach as it pertains to the language network. GLM modeling and gray matter segmentation was performed using SPM12; functional localization was performed using the toolbox released by Ivanova et al. [2020]. Once voxel responses within each brain region were extracted for each trial of each run, some additional preprocessing was required before finalizing the brain representations, and passing them to downstream models. Due to differences in MRI sensitivity across runs, each of the 12 runs of 6 programs for a given subject appeared with nonuniform mean and variance. In order to normalize these signals, so as to leverage data across a participant's entire scanning session, brain representations were z-transformed within each run to achieve a common scale. In order to avoid data-leakage from this preprocessing step, all downstream analyses were cross-validated via a leave-one-run-out approach, such that no intra-run rescaling could be used for prediction.

**MVPA cross-validation and hyperparameters.** For each brain system and each code property or code model, we run a separate MVPA analysis. For each subject, we train a separate ridge regression model on each cross-validated leave-one-run-out fold, and then predict the remaining targets using the brain representations from the left-out run. We use L2 regularization to control for model complexity, and employ a leave-one-out cross validation scheme to select $\lambda$. We score resulting predictions using a series of different metrics to maximize interpretability. For the categorical code properties, we report classification accuracy of model estimates. For the continuously scaled code properties, we report Pearson correlation between the model estimates and true targets. For the model representations, we report rank accuracy between model representation predictions and the true model representations of left-out samples. We calculate rank accuracy as the percentile of a predicted and true target pairing within the full set of possible pairings sorted by a Euclidean distance metric. Rank accuracy was chosen for this experiment as it returns a more interpretable 50% baseline through which to evaluate the brain to model mappings. Following metric calculation on each cross-validation fold, we take the mean of those estimates to report an out-of-sample score over the entire dataset for each subject. We then take the mean of those scores across subjects to derive an overall performance measure. As a baseline, we repeat the above process in its entirety 1000 times for each MVPA analysis, to provide a null permutation distribution against which to compare scores. These null distributions are fit with a univariate Gaussian and used to calculate $z$-statistics for MVPA scores. For each statistical comparison, a single-tail one-sample $z$-test is performed between the MVPA score and the distribution fit to the null permutations. Each of these $p$-values is then FDR-corrected for the set of comparisons in a given experiment, e.g., 4 brain networks $*$ 9 code models = 36 comparisons for experiment 2 (Figure 3). Following FDR-correction, the significance threshold for the resulting $p$-value is set at $alpha = 0.001$, so decoding is deemed significant at $p < 0.001$. The code to run our MVPA analysis, and all resulting statistics, is available in `https://github.com/ALFA-group/code-representations-ml-brain`.

**Code properties.** The code properties we consider are the following –
- ***Token count.*** The total number of tokens that appear in the program. This is a sanity check to measure brain activity against, since it is natural to expect the activity in the brain to be correlated

to the length of a program, and in general, the amount of content being comprehended. By their design, the dataset had program lengths with a small standard deviation.

- **Node count.** The total number of AST nodes that appear in a program. Similar to the number of tokens, we verify if brain activity correlates to a proxy for the amount of syntactic content in it.
- **Runtime steps.** We execute the programs and measure the number of instructions the program steps through. While two programs can have the same program length, the number of instructions executed can differ (*e.g.* `for 1:10` vs. `for 1:50`). We measure if any brain regions capture the number of *mental operations* needed to compute the output of a program.
- **Bytecode ops.** We execute the programs and count the number of bytecode operations performed. This metric should mimic the number of *mental operations* performed needed to compute the output of a program.
- **Cyclomatic complexity.** This metric [McCabe, 1976] is used to measure the general complexity of software systems. It is defined as a function of the number of nodes, edges, and the connected components in a program's control flow graph. While its efficacy as a metric to measure software complexity has been contested [Shepperd, 1988], we include it to see if brain activity is correlated to any explicit syntactic constructs of a program.
- **Halstead difficulty.** This metric [Halstead, 1977] was defined to measure how difficult any piece of software would be for a programmer to comprehend or write. It is defined as a function of the number of tokens, operations, vocabulary that appears in a program.

| **code** | **sentence** |
|---|---|
| height = 5
weight = 100
bmi = weight / (height*height)
print(bmi) | Your height is 5 feet and your weight is 100 pounds. The BMI is defined as the ratio between the weight and the square of the height of a person. What is your BMI? |

Figure 4: An example of a code problem and its sentence equivalent from Ivanova et al. [2020]

The inter-correlations between these metrics have been tabulated in Table 19, Appendix H.

**Code models - Model details.** We evaluate the following models in this work–

- **bag-of-words**, **TF-IDF**. These are count-based language models. They predict the likelihood of a token appearing in a program based on vocabulary statistics and frequencies.
- **seq2seq** [Sutskever et al., 2014], **CodeTransformer** [Zügner et al., 2021], **CodeBERT** [Feng et al., 2020], **CodeBERTa** [HuggingFace, 2020]. We evaluate these three autoencoder (AE) models with increasing model complexity. AE based pretraining reconstructs the original program from corrupted input.
- **XLNet** [Yang et al., 2019], **CodeGPT** [Microsoft, 2021]. We evaluate auto-regressive (AR) models with a model complexity similar to that of the other transformers. AR language modeling estimates the probability distribution of a text corpus in one direction–either forward or backward, while AE models capture dependencies in both directions. We use AR models to mimic a top-down, single pass comprehension style by humans.
- **Token Projection**. We compare the results of the above models against an aggressive baseline provided by using unique Gaussian-distributed random vectors for the token embeddings in a vocabulary, and returning the sum of these token embeddings across a program. The resultant embedding is not transformed by any model or any weights–it instead serves as a proxy for the tokens that appear in the program. This sets a higher bar than the null-distribution labeling baseline (Section 5.2).

**Code models - Configuration details.** In setting up our code model bench, we aimed to select a collection of models ranging in their complexity, namely *bag-of-words*, *TF-IDF* (Term frequency-Inverse document frequency), *seq2seq*, *XLNet*, *CodeTransformer*, *CodeGPT*, *CodeBERT*, and *CodeBERTa*, as well as a *Token Projection* model.

- The *bag-of-words*, *TF-IDF*, *seq2seq*, and *Token Projection* models all use the same custom tokenizer defined by the current authors, whereas *XLNet*, *CodeTransformer*, *CodeGPT*, *CodeBERT*, and *CodeBERTa* use tokenizers defined by the original training authors [Zügner et al., 2021, Microsoft, 2021, Feng et al., 2020, HuggingFace, 2020].

- The tokenizer set up for the models trained by the current authors establishes a unique token in the vocabulary for each Python keyword, each Python builtin function, one token for all numeric types, one token for all string types, and $N$ tokens for each of the $N$ variables in a given program.
- In the case of the *Token Projection* baseline, this tokenizer was used to map each token in a piece of source code to an index in the vocabulary, which in turn was used to index a Gaussian distributed random matrix ($D = 128$). The sum of these random token projections was calculated to return a unique embedding for each program.
- For *bag-of-words* and *TF-IDF*, the validation split of the *code-search-net* Python dataset [Husain et al., 2019] was used to enumerate vocabulary and token occurrence statistics. These data were then used to transform each program to a vector where each dimension represented the raw (*bag-of-words*) or document-weighted (*TF-IDF*) count of the unique items in the vocabulary.
- If a given set of programs viewed by a subject never included a specific token, leading that column to equal the zero vector, then those dimensions would be filtered out.
- We used the dataset released by Project Codenet [Puri et al., 2021] to pretrain a *seq2seq* model [Sutskever et al., 2014]. The dataset contains Python programs that are solutions to olympiad-style programming problems in data structures and algorithms. We trained a seq2seq model with a GRU unit for 15 epochs, with a dropout probability of 0.2, dot product attention, and a maximum sentence length of 500.
- For *XLNet*, *CodeTransformer*, *CodeGPT*, *CodeBERT*, and *CodeBERTa*, each program was passed through that model's tokenizer and model pipeline as defined by the original authors. The representations of each program were extracted from the pretrained encoders. *XLNet*, *CodeTransformer*, and *CodeGPT* were originally trained on the Python subset of the *code-search-net* dataset, whereas *CodeBERT* and *CodeBERTa* were originally trained on the full *code-search-net* dataset, which also incorporates *go*, *java*, *javascript*, *php*, and *ruby*.

# C Additional Results

## C.1 Experiment 1

Using representations from localized brain regions, we attempt to decode static and dynamic properties of comprehended code, and learn maps to code model representations of that same code. We report decoding performance of each brain region to the original Ivanova et al. [2020] code properties in Table 1, the complexity-related code properties in Table 2, and the code model mappings in Table 3. We find that the MD system and the Language system decode all code properties and code models except *variable language* significantly above baseline, as established through a null permutation test. The Visual system decodes 4 of the 6 code properties and 8 of the 9 code models, whereas the Auditory system only decodes the *code vs sentences* property.

| Brain Representation Code Properties | Empirical Baseline | MD | Language | Visual | Auditory |
|---|---|---|---|---|---|
| Code vs. Sentence | 0.55 | 0.88 (+0.33) | 0.87 (+0.32) | 0.88 (+0.33) | 0.64 (+0.09) |
| Control Flow | 0.33 | 0.49 (+0.16) | 0.45 (+0.12) | 0.39 (+0.06) | 0.36 (+0.03) |
| Data Type | 0.49 | 0.63 (+0.14) | 0.58 (+0.09) | 0.61 (+0.12) | 0.53 (+0.04) |
| Variable Language | 0.50 | 0.53 (+0.03) | 0.51 (+0.01) | 0.53 (+0.03) | 0.50 (-0.00) |

Table 1: Brain region decoding performance on original Ivanova et al. [2020] code properties. Scores represent classification accuracy and are contrasted with an empirical baseline from the null permutation analysis. Values in parentheses are units above baseline.

| Brain Representation Code Properties | MD | Language | Visual | Auditory |
|---|---|---|---|---|
| Static Analysis | 0.17 | 0.24 | 0.09 | 0.00 |
| Dynamic Analysis | 0.33 | 0.20 | 0.17 | 0.12 |

Table 2: Brain region decoding performance on *static analysis* and *dynamic analysis* code properties. Scores represent Pearson correlation between predicted and true code properties.

## C.2 Experiment 2

| Brain Representation Code Models | Empirical Baseline | MD | Language | Visual | Auditory |
|---|---|---|---|---|---|
| Token Projection | 0.50 | 0.56 (+0.06) | 0.55 (+0.05) | 0.54 (+0.04) | 0.53 (+0.03) |
| CodeBERTa | 0.50 | 0.60 (+0.10) | 0.57 (+0.07) | 0.58 (+0.08) | 0.53 (+0.03) |
| CodeTransformer | 0.50 | 0.59 (+0.09) | 0.56 (+0.06) | 0.55 (+0.05) | 0.52 (+0.02) |
| CodeBERT | 0.50 | 0.57 (+0.07) | 0.56 (+0.06) | 0.57 (+0.07) | 0.52 (+0.02) |
| CodeGPT | 0.50 | 0.57 (+0.07) | 0.57 (+0.07) | 0.57 (+0.07) | 0.52 (+0.02) |
| XLNet | 0.50 | 0.57 (+0.07) | 0.55 (+0.05) | 0.54 (+0.04) | 0.52 (+0.02) |
| Seq2Seq | 0.50 | 0.57 (+0.07) | 0.54 (+0.04) | 0.51 (+0.01) | 0.51 (+0.01) |
| TF-IDF | 0.50 | 0.57 (+0.07) | 0.54 (+0.04) | 0.53 (+0.03) | 0.52 (+0.02) |
| Bag Of Words | 0.50 | 0.58 (+0.08) | 0.56 (+0.06) | 0.54 (+0.04) | 0.52 (+0.02) |

Table 3: Brain region decoding performance on code model mappings. Scores represent rank accuracy and are contrasted with an empirical baseline from the null permutation analysis. Values in parentheses are units above baseline.

We additionally evaluate whether code models contain linearly decodable information about the code properties we explore in this work (Section 4.2). The models decode with near perfect accuracy on most classification tasks (avg. accuracy$= 0.97 \pm 0.03$), and with high correlations on the regression tasks (avg. $r = 0.89 \pm 0.05$). This is expected for the classification tasks, since the dataset contains tokens which help with perfectly separable decision boundaries. For example, for the *control flow* code property, the dataset contains programs with if, for, or neither, but never both. These results suggest that the code models we evaluate faithfully encode the set of properties we evaluate them on.

| Code Properties Empirical Baseline Model Representation | Control Flow 0.31 | Data Type 0.48 | Dynamic Analysis 0.00 | Static Analysis 0.00 |
|---|---|---|---|---|
| Token Projection | 1.00 | 0.94 | 0.88 | 1.00 |
| CodeBERTa | 0.98 | 1.00 | 0.87 | 0.86 |
| CodeTransformer | 1.00 | 0.95 | 0.90 | 0.91 |
| CodeBERT | 1.00 | 0.99 | 0.89 | 0.92 |
| CodeGPT | 0.98 | 1.00 | 0.84 | 0.82 |
| XLNet | 0.92 | 0.92 | 0.76 | 0.88 |
| Seq2Seq | 0.95 | 0.97 | 0.87 | 0.91 |
| TF-IDF | 1.00 | 0.94 | 0.94 | 0.89 |
| Bag Of Words | 1.00 | 0.92 | 0.93 | 0.98 |

Table 4: Decoding performance of all models on all tasks. Scores represent classification accuracy for *control flow* and *data type*, and Pearson correlation for the remaining benchmarks. These can be contrasted with an empirical baseline from the null permutation analysis.

# D    Analysis of Variance

We run a series of one-way ANOVA statistical tests to assess whether decoding performance varies across brain regions for each code property and code model, and whether decoding performance varies across code models for each brain region. We report these results in Tables 5, 6, and 7. We find that decoding performance varies across brain region for all code properties and code models, except the *variable language* property and the *Token Projection* model, and decoding performance varies across code models for the MD system and the Visual system.

| Code Property | F | p | p (corrected) | Is Significant? |
|---|---|---|---|---|
| Code vs. Sentence | 53.57 | 4.03e-20 | 2.42e-19 | 1 |
| Control Flow | 12.52 | 6.13e-07 | 1.84e-06 | 1 |
| Static Analysis | 6.71 | 3.82e-04 | 7.65e-04 | 1 |
| Data Type | 4.46 | 5.66e-03 | 8.50e-03 | 1 |
| Dynamic Analysis | 4.15 | 8.38e-03 | 1.01e-02 | 1 |
| Variable Language | 0.82 | 4.84e-01 | 4.84e-01 | 0 |

Table 5: Results from statistical testing of variance across brain regions for each code property.

| Code Model | F | p | p (corrected) | Is Significant? |
|---|---|---|---|---|
| Bag Of Words | 5.66 | 1.33e-03 | 1.71e-03 | 1 |
| CodeBERT | 8.73 | 3.72e-05 | 9.75e-05 | 1 |
| CodeBERTa | 12.48 | 6.40e-07 | 5.60e-06 | 1 |
| CodeGPT | 7.00 | 2.73e-04 | 4.09e-04 | 1 |
| CodeTransformer | 11.85 | 1.25e-06 | 5.60e-06 | 1 |
| Seq2Seq | 8.40 | 5.42e-05 | 9.75e-05 | 1 |
| TF-IDF | 8.46 | 5.08e-05 | 9.75e-05 | 1 |
| XLNet | 3.48 | 1.90e-02 | 2.14e-02 | 1 |
| Token Projection | 2.32 | 8.04e-02 | 8.04e-02 | 0 |

Table 6: Results from statistical testing of variance across brain regions for each code model.

| Brain Region | F | p | p (corrected) | Is Significant? |
|---|---|---|---|---|
| Language | 2.49 | 1.35e-02 | 2.70e-02 | 1 |
| Visual | 5.13 | 7.61e-06 | 3.05e-05 | 1 |
| Auditory | 0.30 | 9.66e-01 | 9.66e-01 | 0 |
| MD | 1.79 | 7.97e-02 | 1.06e-01 | 0 |

Table 7: Results from statistical testing of variance across code models for each brain region.

# E   Pairwise Analysis

To extend the findings from the ANOVA analyses towards specific pairwise comparisons between brain regions and code models, we compare scores between brain regions for each code property or code model, and compare scores between code models and a subset of code properties for each brain region, using paired two-tailed $t$-tests. We report all pairwise brain region comparisons in Tables 8 and 10, and report all pairwise code property and model comparisons in Tables 9 and 11. To highlight a few key results from code properties, we find that the MD system decodes *dynamic analysis* significantly better than the LS (Table 8). Moving onto code models, we find that the MD system decodes *CodeBERTa*, *CodeTransformer*, *seq2seq* and *TF-IDF* significantly more accurately than the Language system (Table 10). Finally, an investigation into model complexity reveals that *CodeBERTa*, *CodeTransformer*, and *bag-of-words* are significantly more accurately decoded from the MD system than the *Token Projection* model.

| Code Property | Brain Region A | Brain Region B | t | p | p (corrected) | Is Significant? |
|---|---|---|---|---|---|---|
| Code vs. Sentence | Language | Auditory | 8.21 | 2.74e-08 | 3.29e-07 | 1 |
| Code vs. Sentence | MD | Auditory | 10.42 | 3.52e-10 | 6.33e-09 | 1 |
| Code vs. Sentence | Visual | Auditory | 10.52 | 2.92e-10 | 6.33e-09 | 1 |
| Control Flow | Language | Auditory | 3.90 | 7.20e-04 | 3.24e-03 | 1 |
| Control Flow | MD | Auditory | 4.86 | 6.63e-05 | 5.50e-04 | 1 |
| Control Flow | MD | Visual | 4.22 | 3.26e-04 | 1.96e-03 | 1 |
| Data Type | Language | Auditory | 2.50 | 1.98e-02 | 4.75e-02 | 1 |
| Data Type | MD | Auditory | 2.99 | 6.61e-03 | 2.16e-02 | 1 |
| Data Type | Visual | Auditory | 3.59 | 1.56e-03 | 6.23e-03 | 1 |
| Dynamic Analysis | MD | Auditory | 2.93 | 7.59e-03 | 2.28e-02 | 1 |
| Dynamic Analysis | MD | Language | 2.65 | 1.44e-02 | 3.98e-02 | 1 |
| Dynamic Analysis | MD | Visual | 2.53 | 1.86e-02 | 4.75e-02 | 1 |
| Static Analysis | Language | Auditory | 4.80 | 7.64e-05 | 5.50e-04 | 1 |
| Static Analysis | Language | Visual | 4.06 | 4.89e-04 | 2.52e-03 | 1 |
| Static Analysis | MD | Auditory | 3.35 | 2.80e-03 | 1.01e-02 | 1 |
| Code vs. Sentence | Language | Visual | -0.39 | 6.99e-01 | 7.40e-01 | 0 |
| Code vs. Sentence | MD | Language | 0.59 | 5.64e-01 | 6.34e-01 | 0 |
| Code vs. Sentence | MD | Visual | 0.27 | 7.92e-01 | 7.92e-01 | 0 |
| Control Flow | Language | Visual | 2.43 | 2.34e-02 | 5.26e-02 | 0 |
| Control Flow | MD | Language | 1.75 | 9.36e-02 | 1.87e-01 | 0 |
| Control Flow | Visual | Auditory | 1.44 | 1.62e-01 | 2.57e-01 | 0 |
| Data Type | Language | Visual | -1.03 | 3.15e-01 | 4.36e-01 | 0 |
| Data Type | MD | Language | 1.56 | 1.33e-01 | 2.53e-01 | 0 |
| Data Type | MD | Visual | 0.71 | 4.88e-01 | 5.93e-01 | 0 |
| Dynamic Analysis | Language | Auditory | 1.26 | 2.22e-01 | 3.19e-01 | 0 |
| Dynamic Analysis | Language | Visual | 0.52 | 6.06e-01 | 6.61e-01 | 0 |
| Dynamic Analysis | Visual | Auditory | 0.67 | 5.11e-01 | 5.93e-01 | 0 |
| Static Analysis | MD | Language | -1.45 | 1.61e-01 | 2.57e-01 | 0 |
| Static Analysis | MD | Visual | 1.76 | 9.10e-02 | 1.87e-01 | 0 |
| Static Analysis | Visual | Auditory | 1.44 | 1.64e-01 | 2.57e-01 | 0 |
| Variable Language | Language | Auditory | 0.67 | 5.11e-01 | 5.93e-01 | 0 |
| Variable Language | Language | Visual | -0.94 | 3.55e-01 | 4.56e-01 | 0 |
| Variable Language | MD | Auditory | 1.46 | 1.59e-01 | 2.57e-01 | 0 |
| Variable Language | MD | Language | 0.98 | 3.36e-01 | 4.48e-01 | 0 |
| Variable Language | MD | Visual | 0.32 | 7.49e-01 | 7.70e-01 | 0 |
| Variable Language | Visual | Auditory | 1.35 | 1.89e-01 | 2.84e-01 | 0 |

Table 8: Results from paired two-tailed $t$-tests of brain regions for each code property. $+t$ reflects $A > B$, whereas $-t$ reflects $A < B$.

| Brain Region | Code Property A | Code Property B | t | p | p (corrected) | Is Significant? |
|---|---|---|---|---|---|---|
| MD | Static Analysis | Dynamic Analysis | -2.72 | 1.21e-02 | 4.83e-02 | 1 |
| Auditory | Static Analysis | Dynamic Analysis | -1.96 | 6.27e-02 | 1.25e-01 | 0 |
| Language | Static Analysis | Dynamic Analysis | 0.79 | 4.39e-01 | 4.39e-01 | 0 |
| Visual | Static Analysis | Dynamic Analysis | -1.33 | 1.97e-01 | 2.63e-01 | 0 |

Table 9: Results from paired two-tailed $t$-tests of continuous code metrics across brain regions. Only this subset of properties was selected so as to evaluate scores with a consistent metric and baseline. $+t$ reflects $A > B$, whereas $-t$ reflects $A < B$.

| Code Model | Brain Region A | Brain Region B | t | p | p (corrected) | Is Significant? |
|---|---|---|---|---|---|---|
| Bag Of Words | Language | Auditory | 2.84 | 9.22e-03 | 2.36e-02 | 1 |
| Bag Of Words | MD | Auditory | 4.19 | 3.54e-04 | 2.73e-03 | 1 |
| Bag Of Words | MD | Visual | 2.50 | 1.98e-02 | 4.28e-02 | 1 |
| CodeBERT | Language | Auditory | 3.64 | 1.36e-03 | 4.77e-03 | 1 |
| CodeBERT | MD | Auditory | 3.77 | 9.97e-04 | 4.14e-03 | 1 |
| CodeBERT | Visual | Auditory | 5.59 | 1.09e-05 | 2.95e-04 | 1 |
| CodeBERTa | Language | Auditory | 3.63 | 1.41e-03 | 4.77e-03 | 1 |
| CodeBERTa | MD | Auditory | 4.82 | 7.26e-05 | 9.80e-04 | 1 |
| CodeBERTa | MD | Language | 2.69 | 1.30e-02 | 3.05e-02 | 1 |
| CodeBERTa | Visual | Auditory | 3.66 | 1.30e-03 | 4.77e-03 | 1 |
| CodeGPT | Language | Auditory | 3.89 | 7.41e-04 | 4.00e-03 | 1 |
| CodeGPT | MD | Auditory | 4.07 | 4.73e-04 | 2.84e-03 | 1 |
| CodeGPT | Visual | Auditory | 3.81 | 9.10e-04 | 4.14e-03 | 1 |
| CodeTransformer | Language | Auditory | 4.21 | 3.37e-04 | 2.73e-03 | 1 |
| CodeTransformer | MD | Auditory | 5.31 | 2.17e-05 | 3.91e-04 | 1 |
| CodeTransformer | MD | Language | 2.44 | 2.29e-02 | 4.59e-02 | 1 |
| CodeTransformer | MD | Visual | 3.10 | 5.05e-03 | 1.44e-02 | 1 |
| CodeTransformer | Visual | Auditory | 2.89 | 8.36e-03 | 2.26e-02 | 1 |
| Seq2Seq | MD | Auditory | 4.11 | 4.26e-04 | 2.84e-03 | 1 |
| Seq2Seq | MD | Language | 2.82 | 9.63e-03 | 2.36e-02 | 1 |
| Seq2Seq | MD | Visual | 4.58 | 1.33e-04 | 1.44e-03 | 1 |
| TF-IDF | Language | Auditory | 2.54 | 1.85e-02 | 4.16e-02 | 1 |
| TF-IDF | MD | Auditory | 5.77 | 7.03e-06 | 2.95e-04 | 1 |
| TF-IDF | MD | Language | 3.25 | 3.51e-03 | 1.05e-02 | 1 |
| TF-IDF | MD | Visual | 3.79 | 9.52e-04 | 4.14e-03 | 1 |
| Token Projection | MD | Auditory | 2.46 | 2.20e-02 | 4.56e-02 | 1 |
| XLNet | MD | Auditory | 3.31 | 3.09e-03 | 9.82e-03 | 1 |
| XLNet | MD | Visual | 2.41 | 2.45e-02 | 4.73e-02 | 1 |
| Bag Of Words | Language | Visual | 1.48 | 1.52e-01 | 2.32e-01 | 0 |
| Bag Of Words | MD | Language | 1.51 | 1.44e-01 | 2.29e-01 | 0 |
| Bag Of Words | Visual | Auditory | 0.95 | 3.54e-01 | 4.34e-01 | 0 |
| CodeBERT | Language | Visual | -0.73 | 4.74e-01 | 5.45e-01 | 0 |
| CodeBERT | MD | Language | 0.82 | 4.22e-01 | 4.95e-01 | 0 |
| CodeBERT | MD | Visual | 0.31 | 7.56e-01 | 7.85e-01 | 0 |
| CodeBERTa | Language | Visual | -1.28 | 2.13e-01 | 2.88e-01 | 0 |
| CodeBERTa | MD | Visual | 1.36 | 1.86e-01 | 2.64e-01 | 0 |
| CodeGPT | Language | Visual | 0.06 | 9.53e-01 | 9.53e-01 | 0 |
| CodeGPT | MD | Language | 0.54 | 5.92e-01 | 6.53e-01 | 0 |
| CodeGPT | MD | Visual | 0.63 | 5.38e-01 | 6.05e-01 | 0 |
| CodeTransformer | Language | Visual | 0.88 | 3.86e-01 | 4.64e-01 | 0 |
| Seq2Seq | Language | Auditory | 1.82 | 8.19e-02 | 1.42e-01 | 0 |
| Seq2Seq | Language | Visual | 2.23 | 3.57e-02 | 6.64e-02 | 0 |
| Seq2Seq | Visual | Auditory | 0.14 | 8.91e-01 | 9.08e-01 | 0 |
| TF-IDF | Language | Visual | 1.04 | 3.11e-01 | 3.90e-01 | 0 |
| TF-IDF | Visual | Auditory | 1.34 | 1.94e-01 | 2.69e-01 | 0 |
| Token Projection | Language | Auditory | 1.81 | 8.39e-02 | 1.42e-01 | 0 |
| Token Projection | Language | Visual | 0.37 | 7.15e-01 | 7.72e-01 | 0 |
| Token Projection | MD | Language | 1.09 | 2.85e-01 | 3.66e-01 | 0 |
| Token Projection | MD | Visual | 1.11 | 2.79e-01 | 3.66e-01 | 0 |
| Token Projection | Visual | Auditory | 1.38 | 1.80e-01 | 2.62e-01 | 0 |
| XLNet | Language | Auditory | 1.58 | 1.27e-01 | 2.09e-01 | 0 |
| XLNet | Language | Visual | 0.34 | 7.35e-01 | 7.79e-01 | 0 |
| XLNet | MD | Language | 1.98 | 6.03e-02 | 1.09e-01 | 0 |
| XLNet | Visual | Auditory | 1.47 | 1.55e-01 | 2.32e-01 | 0 |

Table 10: Results from paired two-tailed $t$-tests of brain regions for each code model. $+t$ reflects $A > B$, whereas $-t$ reflects $A < B$.

| Brain Region | Code Model A | Code Model B | t | p | p (corrected) | Is Significant? |
|---|---|---|---|---|---|---|
| Language | CodeBERTa | Seq2Seq | 3.36 | 2.71e-03 | 2.94e-02 | 1 |
| Language | CodeBERTa | TF-IDF | 4.66 | 1.09e-04 | 3.15e-03 | 1 |
| Language | Token Projection | CodeBERTa | -4.16 | 3.83e-04 | 6.89e-03 | 1 |
| MD | Token Projection | Bag Of Words | -3.79 | 9.39e-04 | 1.23e-02 | 1 |
| MD | Token Projection | CodeBERTa | -4.02 | 5.38e-04 | 8.60e-03 | 1 |
| MD | Token Projection | CodeTransformer | -3.21 | 3.92e-03 | 3.53e-02 | 1 |
| Visual | CodeBERT | Seq2Seq | 6.25 | 2.22e-06 | 2.45e-04 | 1 |
| Visual | CodeBERT | TF-IDF | 4.57 | 1.36e-04 | 3.15e-03 | 1 |
| Visual | CodeBERTa | Bag Of Words | 3.83 | 8.57e-04 | 1.23e-02 | 1 |
| Visual | CodeBERTa | CodeTransformer | 3.06 | 5.55e-03 | 4.71e-02 | 1 |
| Visual | CodeBERTa | Seq2Seq | 6.07 | 3.40e-06 | 2.45e-04 | 1 |
| Visual | CodeBERTa | TF-IDF | 4.88 | 6.21e-05 | 2.24e-03 | 1 |
| Visual | CodeBERTa | XLNet | 3.34 | 2.86e-03 | 2.94e-02 | 1 |
| Visual | CodeGPT | Seq2Seq | 4.52 | 1.53e-04 | 3.15e-03 | 1 |
| Visual | CodeGPT | TF-IDF | 3.27 | 3.39e-03 | 3.26e-02 | 1 |
| Visual | CodeTransformer | Seq2Seq | 3.68 | 1.24e-03 | 1.48e-02 | 1 |
| Visual | Token Projection | CodeBERTa | -5.20 | 2.82e-05 | 1.36e-03 | 1 |
| Auditory | CodeBERT | Bag Of Words | -0.37 | 7.17e-01 | 8.96e-01 | 0 |
| Auditory | CodeBERT | CodeGPT | 0.12 | 9.08e-01 | 9.73e-01 | 0 |
| Auditory | CodeBERT | Seq2Seq | 0.80 | 4.33e-01 | 6.93e-01 | 0 |
| Auditory | CodeBERT | TF-IDF | 0.10 | 9.23e-01 | 9.73e-01 | 0 |
| Auditory | CodeBERT | XLNet | -0.50 | 6.22e-01 | 8.54e-01 | 0 |
| Auditory | CodeBERTa | Bag Of Words | 0.14 | 8.94e-01 | 9.73e-01 | 0 |
| Auditory | CodeBERTa | CodeBERT | 0.90 | 3.77e-01 | 6.54e-01 | 0 |
| Auditory | CodeBERTa | CodeGPT | 0.97 | 3.43e-01 | 6.33e-01 | 0 |
| Auditory | CodeBERTa | CodeTransformer | 0.92 | 3.66e-01 | 6.54e-01 | 0 |
| Auditory | CodeBERTa | Seq2Seq | 1.15 | 2.62e-01 | 5.39e-01 | 0 |
| Auditory | CodeBERTa | TF-IDF | 0.60 | 5.56e-01 | 8.01e-01 | 0 |
| Auditory | CodeBERTa | XLNet | 0.20 | 8.41e-01 | 9.44e-01 | 0 |
| Auditory | CodeGPT | Bag Of Words | -0.40 | 6.94e-01 | 8.84e-01 | 0 |
| Auditory | CodeGPT | Seq2Seq | 0.71 | 4.83e-01 | 7.32e-01 | 0 |
| Auditory | CodeGPT | TF-IDF | 0.05 | 9.61e-01 | 9.83e-01 | 0 |
| Auditory | CodeGPT | XLNet | -0.54 | 5.95e-01 | 8.32e-01 | 0 |
| Auditory | CodeTransformer | Bag Of Words | -0.74 | 4.69e-01 | 7.18e-01 | 0 |
| Auditory | CodeTransformer | CodeBERT | -0.40 | 6.90e-01 | 8.84e-01 | 0 |
| Auditory | CodeTransformer | CodeGPT | -0.36 | 7.22e-01 | 8.96e-01 | 0 |
| Auditory | CodeTransformer | Seq2Seq | 0.31 | 7.58e-01 | 9.02e-01 | 0 |
| Auditory | CodeTransformer | TF-IDF | -0.32 | 7.51e-01 | 9.02e-01 | 0 |
| Auditory | CodeTransformer | XLNet | -1.07 | 2.98e-01 | 5.90e-01 | 0 |
| Auditory | Seq2Seq | Bag Of Words | -0.88 | 3.86e-01 | 6.54e-01 | 0 |
| Auditory | Seq2Seq | TF-IDF | -0.58 | 5.65e-01 | 8.06e-01 | 0 |
| Auditory | TF-IDF | Bag Of Words | -0.61 | 5.50e-01 | 8.01e-01 | 0 |
| Auditory | Token Projection | Bag Of Words | 0.16 | 8.77e-01 | 9.64e-01 | 0 |
| Auditory | Token Projection | CodeBERT | 0.47 | 6.39e-01 | 8.60e-01 | 0 |
| Auditory | Token Projection | CodeBERTa | -0.07 | 9.47e-01 | 9.83e-01 | 0 |
| Auditory | Token Projection | CodeGPT | 0.52 | 6.11e-01 | 8.46e-01 | 0 |
| Auditory | Token Projection | CodeTransformer | 0.90 | 3.77e-01 | 6.54e-01 | 0 |
| Auditory | Token Projection | Seq2Seq | 1.01 | 3.21e-01 | 6.08e-01 | 0 |
| Auditory | Token Projection | TF-IDF | 0.75 | 4.60e-01 | 7.15e-01 | 0 |
| Auditory | Token Projection | XLNet | 0.11 | 9.14e-01 | 9.73e-01 | 0 |
| Auditory | XLNet | Bag Of Words | -0.00 | 1.00e+00 | 1.00e+00 | 0 |
| Auditory | XLNet | Seq2Seq | 1.06 | 2.99e-01 | 5.90e-01 | 0 |
| Auditory | XLNet | TF-IDF | 0.69 | 4.98e-01 | 7.43e-01 | 0 |
| Language | CodeBERT | Bag Of Words | 0.33 | 7.44e-01 | 9.01e-01 | 0 |
| Language | CodeBERT | CodeGPT | -0.19 | 8.53e-01 | 9.45e-01 | 0 |
| Language | CodeBERT | Seq2Seq | 2.29 | 3.14e-02 | 1.13e-01 | 0 |
| Language | CodeBERT | TF-IDF | 2.53 | 1.87e-02 | 9.61e-02 | 0 |
| Language | CodeBERT | XLNet | 2.04 | 5.29e-02 | 1.52e-01 | 0 |
| Language | CodeBERTa | Bag Of Words | 1.37 | 1.83e-01 | 4.11e-01 | 0 |
| Language | CodeBERTa | CodeBERT | 1.54 | 1.37e-01 | 3.41e-01 | 0 |
| Language | CodeBERTa | CodeGPT | 0.89 | 3.84e-01 | 6.54e-01 | 0 |
| Language | CodeBERTa | CodeTransformer | 1.41 | 1.72e-01 | 3.99e-01 | 0 |

| | | | | | | |
|---|---|---|---|---|---|---|
| Language | CodeBERTa | XLNet | 2.81 | 9.99e-03 | 7.17e-02 | 0 |
| Language | CodeGPT | Bag Of Words | 0.45 | 6.60e-01 | 8.80e-01 | 0 |
| Language | CodeGPT | Seq2Seq | 2.73 | 1.19e-02 | 7.17e-02 | 0 |
| Language | CodeGPT | TF-IDF | 2.60 | 1.59e-02 | 8.80e-02 | 0 |
| Language | CodeGPT | XLNet | 2.24 | 3.47e-02 | 1.16e-01 | 0 |
| Language | CodeTransformer | Bag Of Words | 0.22 | 8.31e-01 | 9.44e-01 | 0 |
| Language | CodeTransformer | CodeBERT | -0.21 | 8.36e-01 | 9.44e-01 | 0 |
| Language | CodeTransformer | CodeGPT | -0.36 | 7.22e-01 | 8.96e-01 | 0 |
| Language | CodeTransformer | Seq2Seq | 2.46 | 2.17e-02 | 1.04e-01 | 0 |
| Language | CodeTransformer | TF-IDF | 2.63 | 1.50e-02 | 8.61e-02 | 0 |
| Language | CodeTransformer | XLNet | 1.95 | 6.36e-02 | 1.76e-01 | 0 |
| Language | Seq2Seq | Bag Of Words | -2.33 | 2.90e-02 | 1.10e-01 | 0 |
| Language | Seq2Seq | TF-IDF | -0.62 | 5.43e-01 | 7.98e-01 | 0 |
| Language | TF-IDF | Bag Of Words | -2.02 | 5.49e-02 | 1.55e-01 | 0 |
| Language | Token Projection | Bag Of Words | -2.30 | 3.12e-02 | 1.13e-01 | 0 |
| Language | Token Projection | CodeBERT | -2.16 | 4.16e-02 | 1.24e-01 | 0 |
| Language | Token Projection | CodeGPT | -2.19 | 3.85e-02 | 1.22e-01 | 0 |
| Language | Token Projection | CodeTransformer | -2.18 | 3.98e-02 | 1.22e-01 | 0 |
| Language | Token Projection | Seq2Seq | 0.93 | 3.60e-01 | 6.54e-01 | 0 |
| Language | Token Projection | TF-IDF | 0.42 | 6.80e-01 | 8.83e-01 | 0 |
| Language | Token Projection | XLNet | 0.00 | 1.00e+00 | 1.00e+00 | 0 |
| Language | XLNet | Bag Of Words | -1.48 | 1.52e-01 | 3.68e-01 | 0 |
| Language | XLNet | Seq2Seq | 0.84 | 4.10e-01 | 6.70e-01 | 0 |
| Language | XLNet | TF-IDF | 0.35 | 7.30e-01 | 8.97e-01 | 0 |
| MD | CodeBERT | Bag Of Words | -0.43 | 6.68e-01 | 8.83e-01 | 0 |
| MD | CodeBERT | CodeGPT | 0.11 | 9.10e-01 | 9.73e-01 | 0 |
| MD | CodeBERT | Seq2Seq | 0.49 | 6.32e-01 | 8.58e-01 | 0 |
| MD | CodeBERT | TF-IDF | 0.34 | 7.35e-01 | 8.97e-01 | 0 |
| MD | CodeBERT | XLNet | 0.29 | 7.76e-01 | 9.09e-01 | 0 |
| MD | CodeBERTa | Bag Of Words | 1.74 | 9.55e-02 | 2.50e-01 | 0 |
| MD | CodeBERTa | CodeBERT | 2.74 | 1.16e-02 | 7.17e-02 | 0 |
| MD | CodeBERTa | CodeGPT | 2.95 | 7.26e-03 | 5.50e-02 | 0 |
| MD | CodeBERTa | CodeTransformer | 0.85 | 4.03e-01 | 6.70e-01 | 0 |
| MD | CodeBERTa | Seq2Seq | 2.42 | 2.38e-02 | 1.06e-01 | 0 |
| MD | CodeBERTa | TF-IDF | 2.99 | 6.49e-03 | 5.19e-02 | 0 |
| MD | CodeBERTa | XLNet | 2.37 | 2.65e-02 | 1.06e-01 | 0 |
| MD | CodeGPT | Bag Of Words | -0.56 | 5.81e-01 | 8.20e-01 | 0 |
| MD | CodeGPT | Seq2Seq | 0.42 | 6.81e-01 | 8.83e-01 | 0 |
| MD | CodeGPT | TF-IDF | 0.29 | 7.77e-01 | 9.09e-01 | 0 |
| MD | CodeGPT | XLNet | 0.21 | 8.37e-01 | 9.44e-01 | 0 |
| MD | CodeTransformer | Bag Of Words | 1.03 | 3.14e-01 | 6.08e-01 | 0 |
| MD | CodeTransformer | CodeBERT | 1.43 | 1.65e-01 | 3.91e-01 | 0 |
| MD | CodeTransformer | CodeGPT | 2.19 | 3.90e-02 | 1.22e-01 | 0 |
| MD | CodeTransformer | Seq2Seq | 1.87 | 7.45e-02 | 1.99e-01 | 0 |
| MD | CodeTransformer | TF-IDF | 2.36 | 2.73e-02 | 1.06e-01 | 0 |
| MD | CodeTransformer | XLNet | 2.40 | 2.48e-02 | 1.06e-01 | 0 |
| MD | Seq2Seq | Bag Of Words | -0.91 | 3.74e-01 | 6.54e-01 | 0 |
| MD | Seq2Seq | TF-IDF | -0.22 | 8.29e-01 | 9.44e-01 | 0 |
| MD | TF-IDF | Bag Of Words | -1.20 | 2.43e-01 | 5.06e-01 | 0 |
| MD | Token Projection | CodeBERT | -1.28 | 2.14e-01 | 4.60e-01 | 0 |
| MD | Token Projection | CodeGPT | -1.38 | 1.81e-01 | 4.11e-01 | 0 |
| MD | Token Projection | Seq2Seq | -0.77 | 4.50e-01 | 7.12e-01 | 0 |
| MD | Token Projection | TF-IDF | -1.48 | 1.53e-01 | 3.68e-01 | 0 |
| MD | Token Projection | XLNet | -1.21 | 2.39e-01 | 5.06e-01 | 0 |
| MD | XLNet | Bag Of Words | -0.84 | 4.09e-01 | 6.70e-01 | 0 |
| MD | XLNet | Seq2Seq | 0.20 | 8.45e-01 | 9.44e-01 | 0 |
| MD | XLNet | TF-IDF | 0.06 | 9.49e-01 | 9.83e-01 | 0 |
| Visual | CodeBERT | Bag Of Words | 2.39 | 2.53e-02 | 1.06e-01 | 0 |
| Visual | CodeBERT | CodeGPT | 0.68 | 5.00e-01 | 7.43e-01 | 0 |
| Visual | CodeBERT | XLNet | 2.73 | 1.20e-02 | 7.17e-02 | 0 |
| Visual | CodeBERTa | CodeBERT | 1.36 | 1.87e-01 | 4.15e-01 | 0 |
| Visual | CodeBERTa | CodeGPT | 2.45 | 2.24e-02 | 1.04e-01 | 0 |
| Visual | CodeGPT | Bag Of Words | 2.19 | 3.93e-02 | 1.22e-01 | 0 |
| Visual | CodeGPT | XLNet | 2.26 | 3.37e-02 | 1.16e-01 | 0 |
| Visual | CodeTransformer | Bag Of Words | 1.07 | 2.95e-01 | 5.90e-01 | 0 |

| Visual | CodeTransformer | CodeBERT | -1.63 | 1.16e-01 | 2.98e-01 | 0 |
|--------|-----------------|----------|-------|----------|----------|---|
| Visual | CodeTransformer | CodeGPT | -0.98 | 3.35e-01 | 6.27e-01 | 0 |
| Visual | CodeTransformer | TF-IDF | 2.50 | 2.00e-02 | 9.95e-02 | 0 |
| Visual | CodeTransformer | XLNet | 1.57 | 1.30e-01 | 3.28e-01 | 0 |
| Visual | Seq2Seq | Bag Of Words | -2.15 | 4.22e-02 | 1.24e-01 | 0 |
| Visual | Seq2Seq | TF-IDF | -2.39 | 2.57e-02 | 1.06e-01 | 0 |
| Visual | TF-IDF | Bag Of Words | -0.80 | 4.30e-01 | 6.93e-01 | 0 |
| Visual | Token Projection | Bag Of Words | 0.09 | 9.25e-01 | 9.73e-01 | 0 |
| Visual | Token Projection | CodeBERT | -2.73 | 1.18e-02 | 7.17e-02 | 0 |
| Visual | Token Projection | CodeGPT | -2.28 | 3.23e-02 | 1.14e-01 | 0 |
| Visual | Token Projection | CodeTransformer | -1.35 | 1.91e-01 | 4.16e-01 | 0 |
| Visual | Token Projection | Seq2Seq | 2.55 | 1.77e-02 | 9.46e-02 | 0 |
| Visual | Token Projection | TF-IDF | 1.02 | 3.20e-01 | 6.08e-01 | 0 |
| Visual | Token Projection | XLNet | 0.05 | 9.62e-01 | 9.83e-01 | 0 |
| Visual | XLNet | Bag Of Words | 0.02 | 9.83e-01 | 9.97e-01 | 0 |
| Visual | XLNet | Seq2Seq | 1.89 | 7.11e-02 | 1.93e-01 | 0 |
| Visual | XLNet | TF-IDF | 0.75 | 4.62e-01 | 7.15e-01 | 0 |

Table 11: Results from paired two-tailed $t$-tests of code models for each brain region. $+t$ reflects $A > B$, whereas $-t$ reflects $A < B$.

## F Multi-system partial regression analysis: Scores and significance

In order to investigate whether each of the brain systems included contribute unique information towards the decoding tasks, we combine brain representations from different systems in paired combinations and evaluate effects on downstream decoding performance across all experiments. We find that the addition of MD or LS to VS, relative to VS alone, improves downstream decoding of *code vs sentences*. The same effect is observed for the combination of MD and LS compared to either alone. These data suggest that the MD system and the Language system encode unique variance relevant to the decoding of *code vs sentences*, and this is above and beyond the information encoded in the Visual system or each other individually. Additionally, we observe that the addition of MD to VS, relative to VS alone, improves downstream decoding of *control flow*, *data type*, *dynamic analysis*, and *static analysis*. This same effect is observed for the addition of LS to Visual system for *control flow* and *static analysis*. These data suggest that the MD system and the Language system encode unique information above and beyond information encoded in the Visual system for these decoding tasks (Table 15). We repeat this process for Experiment 2, where we find that the addition of MD to VS improves decoding for 7 models, the addition of MD to LS improves decoding for 7 models, and the addition of LS to Visual system improves decoding for all 9 models (Table 16).

| Brain Representation Code Properties | Empirical Baseline | L+V | MD+L | MD+V |
|---|---|---|---|---|
| Code vs. Sentence | 0.56 | 0.94 (+0.38) | 0.94 (+0.38) | 0.93 (+0.37) |
| Control Flow | 0.33 | 0.45 (+0.12) | 0.49 (+0.16) | 0.49 (+0.16) |
| Data Type | 0.50 | 0.62 (+0.12) | 0.63 (+0.13) | 0.68 (+0.18) |
| Variable Language | 0.50 | 0.53 (+0.03) | 0.53 (+0.03) | 0.53 (+0.03) |

Table 12: Multi-system partial regression analysis on original Ivanova et al. [2020] code properties. Scores represent classification accuracy and are contrasted with an empirical baseline from the null permutation analysis. Values in parentheses are units above baseline.

| Brain Representation Code Properties | L+V | MD+L | MD+V |
|---|---|---|---|
| Dynamic Analysis | 0.22 | 0.32 | 0.31 |
| Static Analysis | 0.19 | 0.21 | 0.20 |

Table 13: Multi-system partial regression analysis on static and dynamic code properties. Scores represent Pearson correlation between predicted and true code properties.

| Brain Representation Code Models | Empirical Baseline | L+V | MD+L | MD+V |
|---|---|---|---|---|
| Token Projection | 0.50 | 0.57 (+0.07) | 0.56 (+0.06) | 0.57 (+0.07) |
| CodeBERTa | 0.50 | 0.60 (+0.10) | 0.62 (+0.12) | 0.62 (+0.12) |
| CodeTransformer | 0.50 | 0.58 (+0.08) | 0.60 (+0.10) | 0.60 (+0.10) |
| CodeBERT | 0.50 | 0.60 (+0.10) | 0.59 (+0.09) | 0.59 (+0.09) |
| CodeGPT | 0.50 | 0.59 (+0.09) | 0.58 (+0.08) | 0.58 (+0.08) |
| XLNet | 0.50 | 0.57 (+0.07) | 0.58 (+0.08) | 0.57 (+0.07) |
| Seq2Seq | 0.50 | 0.55 (+0.05) | 0.57 (+0.07) | 0.56 (+0.06) |
| TF-IDF | 0.50 | 0.56 (+0.06) | 0.58 (+0.08) | 0.57 (+0.07) |
| Bag Of Words | 0.50 | 0.57 (+0.07) | 0.59 (+0.09) | 0.59 (+0.09) |

Table 14: Multi-system partial regression analysis on code model mappings. Scores represent rank accuracy and are contrasted with an empirical baseline from the null permutation analysis. Values in parentheses are units above baseline.

| Code Property | Brain Region A | Brain Region B | t | p | p (corrected) | Is Significant? |
|---|---|---|---|---|---|---|
| Code vs. Sentence | L+V | Language | 6.61 | 9.71e-07 | 1.75e-05 | 1 |
| Code vs. Sentence | L+V | Visual | 6.94 | 4.48e-07 | 1.61e-05 | 1 |
| Code vs. Sentence | MD+L | Language | 5.98 | 4.22e-06 | 5.06e-05 | 1 |
| Code vs. Sentence | MD+L | MD | 5.85 | 5.88e-06 | 5.30e-05 | 1 |
| Code vs. Sentence | MD+V | MD | 4.17 | 3.69e-04 | 1.66e-03 | 1 |
| Code vs. Sentence | MD+V | Visual | 4.71 | 9.70e-05 | 5.82e-04 | 1 |
| Control Flow | L+V | Visual | 4.27 | 2.87e-04 | 1.48e-03 | 1 |
| Control Flow | MD+V | Visual | 4.72 | 9.32e-05 | 5.82e-04 | 1 |
| Data Type | MD+V | MD | 2.88 | 8.46e-03 | 2.32e-02 | 1 |
| Data Type | MD+V | Visual | 3.96 | 6.28e-04 | 2.51e-03 | 1 |
| Dynamic Analysis | MD+L | Language | 2.85 | 9.03e-03 | 2.32e-02 | 1 |
| Dynamic Analysis | MD+V | Visual | 2.89 | 8.19e-03 | 2.32e-02 | 1 |
| Static Analysis | L+V | Visual | 3.63 | 1.39e-03 | 5.01e-03 | 1 |
| Static Analysis | MD+V | Visual | 3.23 | 3.73e-03 | 1.22e-02 | 1 |
| Control Flow | L+V | Language | 0.18 | 8.56e-01 | 9.34e-01 | 0 |
| Control Flow | MD+L | Language | 2.22 | 3.65e-02 | 8.22e-02 | 0 |
| Control Flow | MD+L | MD | -0.25 | 8.07e-01 | 9.34e-01 | 0 |
| Control Flow | MD+V | MD | 0.06 | 9.55e-01 | 1.00e+00 | 0 |
| Data Type | L+V | Language | 1.91 | 6.81e-02 | 1.44e-01 | 0 |
| Data Type | L+V | Visual | 0.90 | 3.77e-01 | 5.78e-01 | 0 |
| Data Type | MD+L | Language | 2.32 | 2.97e-02 | 7.12e-02 | 0 |
| Data Type | MD+L | MD | 0.52 | 6.11e-01 | 7.85e-01 | 0 |
| Dynamic Analysis | L+V | Language | 0.42 | 6.80e-01 | 8.17e-01 | 0 |
| Dynamic Analysis | L+V | Visual | 1.81 | 8.34e-02 | 1.67e-01 | 0 |
| Dynamic Analysis | MD+L | MD | -0.18 | 8.57e-01 | 9.34e-01 | 0 |
| Dynamic Analysis | MD+V | MD | -0.60 | 5.52e-01 | 7.36e-01 | 0 |
| Static Analysis | L+V | Language | -1.40 | 1.73e-01 | 3.29e-01 | 0 |
| Static Analysis | MD+L | Language | -0.88 | 3.86e-01 | 5.78e-01 | 0 |
| Static Analysis | MD+L | MD | 1.34 | 1.95e-01 | 3.50e-01 | 0 |
| Static Analysis | MD+V | MD | 1.01 | 3.22e-01 | 5.29e-01 | 0 |
| Variable Language | L+V | Language | 1.01 | 3.24e-01 | 5.29e-01 | 0 |
| Variable Language | L+V | Visual | -0.00 | 1.00e+00 | 1.00e+00 | 0 |
| Variable Language | MD+L | Language | 0.77 | 4.52e-01 | 6.51e-01 | 0 |
| Variable Language | MD+L | MD | -0.68 | 5.02e-01 | 6.95e-01 | 0 |
| Variable Language | MD+V | MD | -0.44 | 6.64e-01 | 8.17e-01 | 0 |
| Variable Language | MD+V | Visual | 0.00 | 1.00e+00 | 1.00e+00 | 0 |

Table 15: Results from paired two-tailed $t$-tests of joint brain systems with their partial components for each code property. $+t$ reflects $A > B$, whereas $-t$ reflects $A < B$.

| Code Model | Brain Region A | Brain Region B | t | p | p (corrected) | Is Significant? |
|---|---|---|---|---|---|---|
| Bag Of Words | L+V | Visual | 4.22 | 3.29e-04 | 1.98e-03 | 1 |
| Bag Of Words | MD+L | Language | 2.62 | 1.52e-02 | 3.15e-02 | 1 |
| Bag Of Words | MD+V | Visual | 4.02 | 5.29e-04 | 2.71e-03 | 1 |
| CodeBERT | L+V | Language | 3.94 | 6.49e-04 | 2.71e-03 | 1 |
| CodeBERT | L+V | Visual | 3.74 | 1.06e-03 | 3.58e-03 | 1 |
| CodeBERT | MD+L | Language | 2.51 | 1.97e-02 | 3.80e-02 | 1 |
| CodeBERT | MD+L | MD | 2.64 | 1.47e-02 | 3.15e-02 | 1 |
| CodeBERTa | L+V | Language | 3.94 | 6.53e-04 | 2.71e-03 | 1 |
| CodeBERTa | L+V | Visual | 2.81 | 9.87e-03 | 2.22e-02 | 1 |
| CodeBERTa | MD+L | Language | 4.26 | 2.98e-04 | 1.98e-03 | 1 |
| CodeBERTa | MD+L | MD | 2.44 | 2.28e-02 | 3.97e-02 | 1 |
| CodeBERTa | MD+V | Visual | 2.93 | 7.58e-03 | 1.79e-02 | 1 |
| CodeGPT | L+V | Language | 2.47 | 2.14e-02 | 3.86e-02 | 1 |
| CodeGPT | L+V | Visual | 3.45 | 2.20e-03 | 6.61e-03 | 1 |
| CodeTransformer | L+V | Language | 2.39 | 2.56e-02 | 4.33e-02 | 1 |
| CodeTransformer | L+V | Visual | 4.25 | 3.04e-04 | 1.98e-03 | 1 |
| CodeTransformer | MD+L | Language | 3.96 | 6.29e-04 | 2.71e-03 | 1 |
| CodeTransformer | MD+V | Visual | 4.60 | 1.27e-04 | 1.45e-03 | 1 |
| Seq2Seq | L+V | Visual | 4.35 | 2.33e-04 | 1.98e-03 | 1 |
| Seq2Seq | MD+L | Language | 2.98 | 6.73e-03 | 1.79e-02 | 1 |
| Seq2Seq | MD+V | Visual | 4.80 | 7.70e-05 | 1.39e-03 | 1 |
| TF-IDF | L+V | Language | 3.10 | 5.11e-03 | 1.45e-02 | 1 |
| TF-IDF | L+V | Visual | 5.57 | 1.14e-05 | 6.16e-04 | 1 |
| TF-IDF | MD+L | Language | 4.95 | 5.23e-05 | 1.39e-03 | 1 |
| TF-IDF | MD+L | MD | 2.54 | 1.82e-02 | 3.65e-02 | 1 |
| TF-IDF | MD+V | Visual | 3.84 | 8.36e-04 | 3.01e-03 | 1 |
| Token Projection | L+V | Language | 2.34 | 2.85e-02 | 4.67e-02 | 1 |
| Token Projection | L+V | Visual | 2.92 | 7.64e-03 | 1.79e-02 | 1 |
| Token Projection | MD+V | Visual | 2.96 | 7.07e-03 | 1.79e-02 | 1 |
| XLNet | L+V | Language | 2.48 | 2.11e-02 | 3.86e-02 | 1 |
| XLNet | L+V | Visual | 4.58 | 1.34e-04 | 1.45e-03 | 1 |
| XLNet | MD+L | Language | 3.66 | 1.30e-03 | 4.13e-03 | 1 |
| XLNet | MD+V | Visual | 3.85 | 8.22e-04 | 3.01e-03 | 1 |
| Bag Of Words | L+V | Language | 0.84 | 4.10e-01 | 4.51e-01 | 0 |
| Bag Of Words | MD+L | MD | 1.56 | 1.32e-01 | 1.70e-01 | 0 |
| Bag Of Words | MD+V | MD | 0.76 | 4.55e-01 | 4.92e-01 | 0 |
| CodeBERT | MD+V | MD | 2.23 | 3.55e-02 | 5.48e-02 | 0 |
| CodeBERT | MD+V | Visual | 2.25 | 3.40e-02 | 5.40e-02 | 0 |
| CodeBERTa | MD+V | MD | 2.18 | 3.95e-02 | 5.92e-02 | 0 |
| CodeGPT | MD+L | Language | 1.58 | 1.27e-01 | 1.67e-01 | 0 |
| CodeGPT | MD+L | MD | 1.95 | 6.36e-02 | 9.03e-02 | 0 |
| CodeGPT | MD+V | MD | 1.13 | 2.71e-01 | 3.18e-01 | 0 |
| CodeGPT | MD+V | Visual | 2.03 | 5.46e-02 | 7.97e-02 | 0 |
| CodeTransformer | MD+L | MD | 0.98 | 3.38e-01 | 3.83e-01 | 0 |
| CodeTransformer | MD+V | MD | 1.34 | 1.94e-01 | 2.33e-01 | 0 |
| Seq2Seq | L+V | Language | 1.51 | 1.45e-01 | 1.82e-01 | 0 |
| Seq2Seq | MD+L | MD | 0.46 | 6.49e-01 | 6.49e-01 | 0 |
| Seq2Seq | MD+V | MD | -0.69 | 4.97e-01 | 5.26e-01 | 0 |
| TF-IDF | MD+V | MD | 0.49 | 6.30e-01 | 6.42e-01 | 0 |
| Token Projection | MD+L | Language | 1.79 | 8.59e-02 | 1.19e-01 | 0 |
| Token Projection | MD+L | MD | 0.97 | 3.41e-01 | 3.83e-01 | 0 |
| Token Projection | MD+V | MD | 1.38 | 1.80e-01 | 2.20e-01 | 0 |
| XLNet | MD+L | MD | 1.65 | 1.13e-01 | 1.52e-01 | 0 |
| XLNet | MD+V | MD | 0.52 | 6.06e-01 | 6.29e-01 | 0 |

Table 16: Results from paired two-tailed $t$-tests of joint brain systems with their partial components for each code model. $+t$ reflects $A > B$, whereas $-t$ reflects $A < B$.

# G   Ranked correlations of brain and model decoding performance

|                  | MD  | Language | Visual | Auditory |
|------------------|-----|----------|--------|----------|
| CodeBERTa        | **1.0** | 0.4      | 0.4    | 0.2      |
| CodeTransformer  | **1.0** | 0.4      | 0.4    | 0.2      |
| CodeBERT         | **1.0** | 0.4      | 0.4    | 0.2      |
| CodeGPT          | 0.8 | 0.8      | 0.0    | -0.4     |
| XLNet            | 0.8 | 0.8      | 0.0    | -0.4     |
| Seq2Seq          | 0.8 | 0.8      | 0.0    | -0.4     |
| TF-IDF           | **1.0** | 0.4      | 0.4    | 0.2      |
| Bag Of Words     | 0.4 | **1.0**  | -0.6   | -0.8     |
| Token Projection | 0.4 | **1.0**  | -0.6   | -0.8     |

Table 17: Spearman ranked correlations between brain region and code models over the performance of their representations in decoding core code properties. Perfect correspondence is bolded.

We investigate whether there exists any trends in the correspondence between how well each brain region and each code model performs on each decoding task.

For each brain region and code model, we extract the $z$-score of how well each hand-selected property is decoded. We then compare the performances of each brain region to each code model via a Spearman rank correlation.

Under such a framework, if two representations have similar content, and perform similarly in how well or how poorly they decode a battery of properties, they will show strong positive rank correlations. On the other hand, if two representations show different patterns in how well they perform on the same tasks, they will be less strongly correlated. This bears resemblance to Schrimpf et al. [2020b], where pairwise correlations between tasks were explored over models, but here we adopt a complementary approach where models and regions are pairwise correlated over tasks.

Here, we find that the MD system shows perfect task performance rank correlation with 3 transformer architectures: *CodeBERTa*, *CodeBERT*, and *CodeTransformer*, as well as *TF-IDF* (which contains broader distributional information). On the other hand, the Language system is consistent with *bag-of-words* and *Token Projection*, which only reflect the presence of specific tokens.

These data, in conjunction with the core analyses summarized in Section 5 and Appendix C, provide further evidence for the role of the LS in representing the presence of specific tokens, and the MD in representing higher-level content.

# H  All code properties: Scores and Correlations

We analyzed a series of code properties as part of the static and dynamic code analysis. As several of these properties (*token count*, *node count*, *Halstead difficulty*, *cyclomatic complexity*, and *bytecode ops*) were revealed to be highly correlated in a post-hoc analysis, we report only one measure for this subset in Experiment 1, *token count*, but include all scores here for completeness (Table 18). We also report the correlation matrix between all code properties that led us to select *token count* as the representative property for this subset (Table 19).

| Brain Representation Code Properties | MD | Language | Visual | Auditory |
|---|---|---|---|---|
| Token Count | 0.17 | 0.24 | 0.09 | 0.00 |
| Node Count | 0.12 | 0.20 | 0.03 | 0.01 |
| Halstead Difficulty | 0.11 | 0.17 | 0.10 | -0.01 |
| Cyclomatic Complexity | 0.18 | 0.24 | 0.10 | 0.01 |
| Bytecode Operations | 0.15 | 0.18 | 0.03 | 0.02 |
| Runtime Steps | 0.33 | 0.20 | 0.17 | 0.12 |

Table 18: Brain region decoding performance on all *static analysis* and *dynamic analysis* code properties. Scores represent Pearson correlation between predicted and true code properties.

| | Datatype | Conditional | Iteration | Tokens | Nodes | Halstead | Cyclomatic | Runtime | Bytecode |
|---|---|---|---|---|---|---|---|---|---|
| Datatype | 1.00 | 0.00 | 0.00 | 0.41 | 0.33 | 0.39 | 0.18 | 0.13 | 0.28 |
| Conditional | - | 1.00 | -0.50 | 0.44 | 0.46 | 0.50 | 0.61 | -0.41 | 0.49 |
| Iteration | - | - | 1.00 | -0.10 | -0.14 | -0.35 | -0.09 | 0.90 | 0.01 |
| Tokens | - | - | - | 1.00 | 0.97 | 0.89 | 0.80 | 0.08 | 0.95 |
| Nodes | - | - | - | - | 1.00 | 0.86 | 0.79 | 0.00 | 0.96 |
| Halstead | - | - | - | - | - | 1.00 | 0.74 | -0.17 | 0.80 |
| Cyclomatic | - | - | - | - | - | - | 1.00 | 0.02 | 0.84 |
| Runtime | - | - | - | - | - | - | - | 1.00 | 0.13 |
| Bytecode | - | - | - | - | - | - | - | - | 1.00 |

Table 19: Correlation matrix across all code properties. The *control flow* property was split into two binary properties here, *conditional* and *iteration*. Of relevance is that *token count* presents with $r > 0.8$ for all *static analysis* properties in the set and as such is used as the representative *static analysis* code property in Experiment 1.

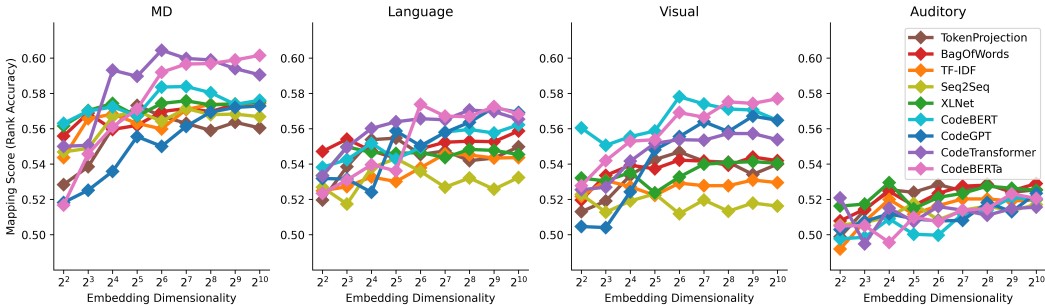

Figure 5: Sensitivity of brain representation mapping to model output dimensions. Each subplot contains the decoding results from a given brain network, and each line reflects a unique code model across a range of controlled embedding dimensions.

# I Sensitivity of brain mapping to model output dimensions

Of potential interest to the decoding framework is not just the complexity of code models, but their intrinsic dimensionality as well. In order to investigate to what extent brain model mappings are robust to changes in code model dimensionality, and to assess which model representations are most sensitive to compression and expansion, we rerun our current decoding framework while controlling for embedding size.

For each brain network to code model mapping task, prior to the MVPA analysis, we control for the dimensionality of the code model embedding via projection through a Gaussian random matrix $\mathbb{R}^{d_1 \times d_2}$ drawn from $N(0, 1/d_2)$ where $d_1$ is the original code model embedding dimensionality and $d_2$ is the desired dimensionality.

We observe that, on average, models of lower complexity (e.g., *bag-of-words*, *TF-IDF*, *seq2seq*) appear relatively robust to compression, whereas the most complex models (e.g., *CodeTransformer*, *CodeGPT*, *CodeBERTa*) gain ample performance from higher dimensional expression, and suffer considerably when constrained.

These data suggest that higher dimensions of the encoder in complex models encode relevant neural information. Additionally, these effects appear to be most pronounced when decoding from the brain region whose representations yield the strongest mappings, the MD system.

We note here however that we could be observing an interaction effect between model complexity and dimensionality output in these results. Since we cannot fully control for the complexity of these models (by making them all 'equally complex'), this experiment alone cannot drive definitive conclusions.

These results instead constitute a preliminary exploration into the effects of code model dimensionality on brain to model representation mappings, and suggest an avenue for future investigation.

## J Robustness of Results to Regression Metric

For the decoding analysis of the continuous valued *dynamic analysis* and *static analysis* properties, it is reasonable to ask why the Pearson correlation metric was chosen as opposed to $RMSE$, as is typically customary for regression tasks. While we present the results using the Pearson correlation metric in the core results for interpretability via the zero-baseline, here we confirm that the use of an $RMSE$ metric leads to the same conclusions. As we see here, the MD, LS, and Visual system decode the *dynamic analysis* property, and MD and LS decode the *static analysis* property with significantly lower $RMSE$ than the null permutation baseline. These results precisely confirm and mirror the patterns observed in Figure 3.

| Brain Network | Code Property | RMSE | Null RMSE | Is Significant? |
|---|---|---|---|---|
| MD | Dynamic Analysis | 3.49 | 4.03 ± 0.08 | 1 |
| MD | Static Analysis | 7.68 | 8.27 ± 0.15 | 1 |
| Language | Dynamic Analysis | 3.68 | 3.95 ± 0.08 | 1 |
| Language | Static Analysis | 7.34 | 8.00 ± 0.15 | 1 |
| Visual | Dynamic Analysis | 3.79 | 4.05 ± 0.08 | 1 |
| Visual | Static Analysis | 7.99 | 8.28 ± 0.16 | 0 |
| Auditory | Dynamic Analysis | 3.79 | 3.98 ± 0.07 | 0 |
| Auditory | Static Analysis | 8.04 | 8.10 ± 0.15 | 0 |

Table 20: $RMSE$ between observed and predicted continuous-valued code properties from the MVPA regression task in Experiment 1, confirming the pattern of results observed in 5.1