# OpenReview forum: "Convergent Representations of Computer Programs in Human and Artificial Neural Networks"
_NeurIPS.cc/2022/Conference — NeurIPS 2022 Accept_

### Official Review · Reviewer_zm5y · 2022-07-10

**Rating:** 6
**Confidence:** 5
**Soundness:** 4 excellent
**Presentation:** 3 good
**Contribution:** 3 good

**Summary:**

This work examines the relationship between fMRI recordings of people who read short programs and different properties and representations of the programming code. The aim of the work is to understand what properties of code are encoded by different brain systems, and to understand how similar the representations of code in the brain are to those encoded by self-supervised language models that are pretrained to encode programming code. The authors find that several program properties can be significantly decoded from 2 brain systems (the multiple demand system and the language system). They further find that representations of the programs extracted from several machine learning models of varying complexity can also be significantly related to these brain systems.

**Questions:**

I think this work is a solid contribution. I do have some questions that I am curious about.
1. I'd appreciate the author's response to the Weakness I raised above.
2. L65-L74 try to motivate why the question of investigating code comprehension is of interest. One reason given is that it's poorly understood, which is fine. The other reason is difficult for me to understand -- the authors state that two previous works have found that the neural bases of code comprehension are different from those of natural language comprehension. And then they ask "do code models mimic human cognition of programs? Do language models?". It's not clear what the difference between code and language models is for the purposes of this work -- aren't all code models just language models that are trained on code rather than natural language, and don't the authors *only* investigate representations from code models and not from natural language models? It's not clear what the expectation here is. Much previous work, as discussed by the authors, shows that language models predict large part of cortex (including the language and multi-demand regions) during natural language comprehension. On the other hand, work on code comprehension shows that its neural bases are different from those of natural language comprehension. So then what did the authors expect to learn by using representations from language models to predict activity during code comprehension? Were these expectations met? Addressing this would strengthen the motivation.
3. L377-380: An intriguing discussion of how the current findings that the language system encodes some properties of code can be understood alongside the findings from previous work that the language system does not show a consistent response to code. If the language system indeed does not show consistent response to code and yet it encodes important code properties, what does this imply about these system-based analyses in general? The whole reason to look at these specific systems is that they are believed to have some function, based on work that examines their average responses. Perhaps a whole-brain analysis would be more telling of the functional organization of code comprehension w.r.t. different stimulus properties.




Minor comments:
- The language "MD/LS network decodes ..." sounds strange to me, and it's used throughout the paper. I would suggest using "encodes" or "[FEATURE] is decodable from MD/LS network"
- L313: "These findings establish the unique role of the MD system in encoding code-simulation and execution related information" -- this should be reworded because it can be misinterpreted as these properties being uniquely encoded in the MD system, which is not the case (as the authors show).
- Typo: L314 "an important of aspect of code" -> "an important aspect of code"
- Table 11 in the appendix is illegible


**Limitations:**

Limitations are sufficiently discussed

**Strengths And Weaknesses:**

Disclaimer: I have reviewed this work previously. The authors have addressed almost all of my previous concerns and the work is much improved. I will focus on the few remaining concerns that I have in this review. I am leaning towards acceptance because of the thorough experimentation and improved discussion and motivation.

Strengths:
- The discussion of related literature is thorough
- The analyses are well motivated and the interpretation of the results is sound
- The writing is clear and concise

Weaknesses:
- There is a disconnect between the two main parts of the paper -- decoding hand-engineered code features from the brain, and decoding representations from code models from the brain, and it currently feels like it's two projects put into one manuscript. The authors motivate well why studying both is important, but it's not clear what is learned from the code models over what can be learned from the hand-engineered code features. I agree that in *theory* investigating code model representations may give us additional insight because they may contain additional information, but that is not shown in the current work, as far as I can see. An improvement would be to show that certain parts of the brain can be predicted better using the code properties than with all of the hand engineered features, via an encoding analysis, and ideally show that code models can explain additional variance in the brain recordings. This will actually show that the code models contain additional brain-relevant information over what is encoded by the hand engineered features.

---

> ### Author Response · Authors · 2022-08-01
> **Response to Reviewer zm5y - part 2**
>
>
> > 3. If the language system indeed does not show consistent response to code and yet it encodes important code properties, what does this imply about these system-based analyses in general?
>
> That’s a great question! In Liu et al. (2020) and Ivanova et al. (2020), the authors did not find significant _aggregate activation_ in the language system (LS) in response to _code comprehension_.
>
> Liu and Ivanova studied whether the response to code comprehension seen in the LS was as strong as responses to text comprehension in the LS. While the response is not as strong in code comprehension, it does not exclude a subset of code-related information being encoded in whatever little response that does show up, which is exactly what our work helps establish.
>
> We show that the LS seems to encode information which is correlated only with token-level information in programs (see response to question 1 above for details), while the MD system encodes dynamic analysis-based properties + properties which complex code model architectures like transformers encode.
>
> We will add this observation to our discussion, in section 6.
>
> ----------------
>
> > 4. The language "MD/LS network decodes ..." sounds strange to me
>
> We have stuck to the language that is generally used in neuroimaging/decoding studies.
>
> “MD/LS network decodes [feature]..” can be read as “representations in the MD/LS network decode [feature]”, which is the same as “[feature] is decodable from the MD/LS network”.
>
> This verbiage also does justice to the decoding pipeline, wherein the input to the linear models are the brain network representations, from which we decode code properties/code model representations. This directionality is relevant, as an encoding analysis leads to a different interpretation.
>
> ----------------
>
> > 5. L313: "These findings establish the unique role of the MD system in encoding code-simulation and execution related information" -- this should be reworded
>
> Thanks! We’ve fixed this.
>
> ----------------
>
> > 6. Table 11 in the appendix is illegible
>
> Thanks, we’ve fixed this now.

---

> > ### Comment · Reviewer_zm5y · 2022-08-05
> > **misunderstanding question**
> >
> > Thanks for the responses. I believe Q3 was misunderstood. I understand how it's possible to decode a property from a brain region that doesn't show an aggregate selective response. My question was, given this finding, isn't an analysis which considers more brain regions (possibly a full brain ROI atlas) more informative? The point is that the investigated systems were chosen because of previous work that considers aggregate responses, but it is very possible that additional regions participate in the encoding of the code properties that are not selective in the aggregate.

---

> > > ### Author Response · Authors · 2022-08-08
> > > **Thanks**
> > >
> > > Thanks for the clarification.
> > >
> > > Yes, there is indeed a possibility that regions not considered here, which did not show aggregate responses in Ivanova et al., might still represent decodable information about code properties.
> > >
> > > The reason that we specifically analyze MD and LS in our work, rather than analyzing an anatomical atlas, is so that we could evaluate functionally rather than anatomically localized networks.
> > >
> > > The dataset from Ivanova et al. included localizer runs for the language and MD networks, which allowed us to precisely outline each functional network uniquely in each participant.
> > >
> > > If we were to have included all ROIs in an anatomical atlas, we would not have been able to draw inferences about the functional profile of the identified regions.

---

> > > > ### Comment · Reviewer_zm5y · 2022-08-09
> > > > **thanks for the response**
> > > >
> > > > Thanks, makes sense. I wonder how important the connection to the functional profile of the regions is at the end of the day anyway, but I'm satisfied with your response in the context of this work.

---

> ### Author Response · Authors · 2022-08-01
> **Response to Reviewer zm5y - part 1**
>
> Thank you for your careful review of our work; we appreciate the detailed feedback you have provided.
>
> We are glad that you found the analyses well-motivated, and the writing clear and concise.
> We address your questions below.
>
> **Answers to your questions**
>
> > 1. There is a disconnect between the two main parts of the paper -- decoding hand-engineered code features from the brain, and decoding representations from code models from the brain, and it currently feels like it's two projects put into one manuscript.
> Thanks for this feedback.
>
> The overall theme was to investigate what code properties can be decoded from brain recordings. We start with hand-engineered features, and then move onto evaluating deep-learning based features, exploring the possibility of uncovering features which cannot be explicitly hand-coded.
>
> We summarize our results below from each of these two experiments to help see how they connect to the overall theme. We would greatly appreciate some more pointed feedback which can identify how this perceived disconnect manifests in the submitted draft, so that this narrative is improved.
>
> **Summary**
>
> - _Brain regions represent code properties._ (Section 5.1 in submitted draft)
>
>     - The MD and LS encode properties about comprehended code, including `control flow`, `datatype`, `static analysis`, and `dynamic analysis`.
>     - The MD decodes `dynamic analysis` properties $>$ `static analysis`
>     - The `dynamic analysis` property is decoded from MD $>$ LS
>
> - _Brain region representations linearly map to code model representations._ (Section 5.2 in submitted draft)
>
>     - The MD and LS linearly map to a collection of code models, spanning from a simple `TokenProjection` to several transformer architectures.
>     - The MD maps to the representations of 3 models more effectively than to the `TokenProjection` model, lending evidence that the MD encodes information beyond the presence of specific tokens. We do not find such evidence for the LS.
>
> In addition to this, we also analyzed another aspect of this data, revealing yet another trend which confirms the unique role of the MD system -
>
> - _Code models and brain regions show correspondence in the properties that can be most-effectively decoded from them._ **(new analysis; added to section 5.2 and Appendix G in the revised draft)**
>
>     - The presented code models encode all the code properties we investigate.
>     - Variation in how well each of these properties are decoded from each model aligns with patterns in how well each of these properties are decoded from specific brain regions.
>     - In particular, the ranked performance of the MD system in decoding these properties is matched perfectly by several complex transformer-based models, whereas the ranked performance of the LS in decoding these properties is matched perfectly by simpler, token-based models like `bag-of-words` and `TokenProjection`.
>
> We hence conclude that the MD system is distinctly involved in encoding properties like dynamic analysis and those properties captured by complex model architectures like transformers.
>
> ----------------
>
> > 2. It's not clear what the difference between code and language models is for the purposes of this work
>
> Thanks for this catch. Yes, the question “Do language models?” can be omitted.
>
> We intended to ask whether representations learned by code models, which are essentially language models in the objectives they are trained on, correspond to the representations in the brain. If we find no correspondence in either of the two primary regions we investigate (MD, LS), then it is worth questioning and reconsidering the objectives we employ to train neural code models.
> However, in our results, we do find a correspondence between different brain regions and code/language model classes (Section 5.2).
>
> ----------------

---

> > ### Comment · Reviewer_zm5y · 2022-08-05
> > **Thanks for the additional experiment**
> >
> > Thanks for the response and the additional experiment that decodes code properties from the code models and compares the ranked performance of the brain systems in decoding these properties to that of the code models. This analysis is in the right direction for connecting the two parts of the paper, but is not all the way there. The reason is that it's still not clear what we gain from the code model comparisons that we didn't already know from the code property decoding experiments. To me, the motivation of just wanting to check whether the code model representations align with those in the brain is not sufficient, because of the already existing work in natural language comprehension that uses essentially the same models (trained on different data) and shows a significant correspondence with the same set of brain regions. So a correspondence between code models and brain representations in the MD and LS systems was highly expected.
> >
> > I consider the analysis that I suggested in my original review a more convincing one:
> > "An improvement would be to show that certain parts of the brain can be predicted better using the code [model representations] than with all of the hand engineered features, via an encoding analysis, and ideally show that code models can explain additional variance in the brain recordings. This will actually show that the code models contain additional brain-relevant information over what is encoded by the hand engineered features."
> > Showing this would also be helpful for the narrative of the manuscript -- it motivates why investigating the relationship with code models is important.

---

> > > ### Author Response · Authors · 2022-08-08
> > > **Thanks**
> > >
> > > Thanks for the clarification, and engaging with us during this discussion period.
> > >
> > > > the motivation of just wanting to check whether the code model representations align with those in the brain is not sufficient
> > >
> > > Yes, however, we do want to evaluate this alignment in the context of what properties code model representations encode.
> > >
> > > While an alignment with the LS and MD was expected, we did not expect a preference for the MD to encode dynamic analysis-related (and more, from the alignment of complex code models) properties, while the LS to encode static properties.
> > >
> > > We do agree an encoding analysis will provide more perspective to this problem.
> > > This is a direction for future work that we are currently pursuing.

---

> > > > ### Comment · Reviewer_zm5y · 2022-08-09
> > > > **continued confusion**
> > > >
> > > > I feel like we're either miscommunicating or the authors are trying to claim something that doesn't actually align with my understanding of the work.
> > > >
> > > > >Yes, however, we do want to evaluate this alignment in the context of what properties code model representations encode.
> > > >
> > > > What part of your work actually evaluates the alignment between code models and the brain in the context of what properties code model representation encode? I don't believe that this work actually does this. I believe this work separately investigates the 3 pairwise relationships: code models and brains, code properties and brains, and code models and code properties.
> > > >
> > > > I feel strongly that the authors should be upfront about this both in this response and in the paper and acknowledge that even though some of their latest experiments are more in the direction of trying to relate all three, they are definitely not direct evidence.

---

> > > > > ### Author Response · Authors · 2022-08-09
> > > > > **We were referring to our most recent analysis**
> > > > >
> > > > > We're sorry if this has been confusing, and thank you for making these clarifications.
> > > > >
> > > > > > What part of your work actually evaluates the alignment between code models and the brain in the context of what properties code model representation encode?
> > > > >
> > > > > We were referring to the most recent analysis done during the rebuttal period (appendix G), in which we found the ranked performance of the MD system in decoding code properties is matched perfectly by several complex transformer-based models, while in the LS, it is matched perfectly by token-based models.
> > > > >
> > > > > Yes, we do analyze the 3 pairwise relationships that you mention, and make the additional ranked analysis.
> > > > >
> > > > > And we agree--while the ranked analysis is not direct evidence for tying them all together, they are encouraging initial results.
> > > > >
> > > > > We do not make any stronger claims than these, and to avoid any miscommunication, we will ensure our verbiage faithfully represents the work done.

---

> > > > > > ### Comment · Reviewer_zm5y · 2022-08-09
> > > > > > **resolved**
> > > > > >
> > > > > > Great, sounds like we are on the same page. Please also make sure to make this disclaimer in the paper itself as well.

---

> > > > > > > ### Author Response · Authors · 2022-08-09
> > > > > > > **thanks**
> > > > > > >
> > > > > > > Thanks again! Will do.

---

### Official Review · Reviewer_6vc3 · 2022-07-11

**Rating:** 6
**Confidence:** 2
**Soundness:** 3 good
**Presentation:** 2 fair
**Contribution:** 2 fair

**Summary:**


This paper studies the parallels in the representation of computer code comprehension, both in neural activity as it is recorded in fMRI and in an artificial intelligence system. The paper first studies the state of the art and presents the scientific question by describing the different aspects of computer code comprehension. It then presents the different hypotheses that can be made at the level of the brain organization and how the understanding of a computer code can be assigned to certain regions of the cortex. The paper then proposes an artificial intelligence architecture, of the deep learning type, which allows building a model of code representation that is evaluated in two experiments that are those described in an existing dataset. The results show a similarity between the results obtained in the neural data and the artificial model.


**Questions:**


I have a question about the paper: does a person who knows how to code in several languages (Python, C++, ...) acquire representations specific to each property of the code that are superimposed in the brain as well as in the artificial system, in the same way as people who are polyglot?


**Limitations:**


The authors have correctly specified the limitations (but no potentially negative societal impacts) of this paper.


**Strengths And Weaknesses:**


The main strength of the paper is the originality of the study and its potential societal impact in terms of better understanding of the code that can be performed by computer scientists. The analyses seem correct, even if I am not a specialist in the field. The main weakness of the paper is to understand to what extent the comparison results obtained in figure 3 can be interpreted as a homology between the processing that is actually done in the brain and the one represented in the artificial system. (minor) Note that in figure 2, the code that is shown as an illustration does not seem valid.

---

> ### Author Response · Authors · 2022-08-01
> **Response to Reviewer 6vc3 - part 2**
>
> > 2. does a person who knows how to code in several languages (Python, C++, ...) acquire representations specific to each property of the code, similar to polyglots.
>
> That’s a great question. We are unsure, and understanding how programs from different languages are represented in the LS/MD is a compelling direction for future work.
>
> This is our current understanding about languages and polyglots:
> - The language system (LS) consistently and uniquely responds to language utterances irrespective of the family a language belongs to (Indo-European, Afroasiatic, etc.).
>     - “An investigation across 45 languages and 12 language families reveals a universal language network” by Malik-Moraleda et al., 2022
>     - The language system is exclusively involved in core linguistic processes such as lexical access or composition (Fedorenko and Shain, 2021).
> - There exists a correspondence between representations of words/sentences across multiple languages.
>     - “Identifying bilingual semantic neural representations across languages”, by Buchweitz et al., 2012
>     - “Commonalities and differences in the neural representations of English, Portuguese, and Mandarin sentences: When knowledge of the brain-language mappings for two languages is better than one” by Yang et al., 2017
>    - “Similarities and differences in the neural representations of abstract concepts across English and Mandarin” by Vargas et al., 2022
> - There is preliminary evidence showing that semantic similarity of words/sentences (e.g. action verbs like fetch, grasp, etc.) are encoded by a range of cortical regions.
>     - “Representational Similarity Mapping of Distributional Semantics in Left Inferior Frontal, Middle Temporal, and Motor Cortex” by Carota et al., 2017
> - There’s similar evidence in the motor and speech cortices as well.
>     - “Hierarchical Organization of Auditory and Motor Representations in Speech Perception: Evidence from Searchlight Similarity Analysis” by Evans et al., 2015
>     - “Imagined and Executed Actions in the Human Motor System: Testing Neural Similarity Between Execution and Imagery of Actions with a Multivariate Approach”, by Zabicki et al., 2017
>
> These are just initial investigations made in language research, leaving a lot more to be investigated thoroughly.
> Just a small fraction of these questions have been investigated for programming languages though.
>
> Ivanova et al. (2020) analyzed two languages–Python and Scratch Jr. and found a consistent overall activation in the MD but not in the LS.
> We could not analyze programs written in Scratch Jr. from their dataset since programs in Python did not correspond to those in Scratch Jr.
>
> We are in the process of collecting more data on more languages to investigate the line of questions you inquire about in addition to other similar questions.
>
> ----------------
>
> > 3. Minor comment on Figure 2.
>
> Thanks for this note.
> The code is just a stock image of a Python implementation, and as you rightly point out, is an incomplete implementation to pretty print the AST of a Python program.
> We will change it to something simpler.
>
> ----------------

---

> ### Author Response · Authors · 2022-08-01
> **Response to Reviewer 6vc3 - part 1**
>
> Thank you for your review of this work, and we are glad that you found this work of high impact to computer scientists interested in better understanding the cognitive aspects of programming.
>
> We address specific questions and comments below.
>
> > 1. To what extent can comparing results obtained in figure 3 be interpreted as a homology between the processing that is actually done in the brain and the one represented in the artificial system.
>
> This is an important question. We have the following evidence:
>
>
> - _Brain regions represent code properties._ (Section 5.1 in submitted draft)
>
>     - The MD and LS encode properties about comprehended code, including `control flow`, `datatype`, `static analysis`, and `dynamic analysis`.
>     - The MD decodes `dynamic analysis` properties $>$ `static analysis`
>     - The `dynamic analysis` property is decoded from MD $>$ LS
>
> - _Brain region representations linearly map to code model representations._ (Section 5.2 in submitted draft)
>
>     - The MD and LS linearly map to a collection of code models, spanning from a simple `TokenProjection` to several transformer architectures.
>     - The MD maps to the representations of 3 models more effectively than to the `TokenProjection` model, lending evidence that the MD encodes information beyond the presence of specific tokens. We do not find such evidence for the LS.
>
> In addition to this, we also analyzed another aspect of this data, revealing yet another trend which confirms the unique role of the MD system -
>
> - _Code models and brain regions show correspondence in the properties that can be most-effectively decoded from them._ **(new analysis; added to section 5.2 and Appendix G in the revised draft)**
>
>     - The presented code models encode all the code properties we investigate.
>     - Variation in how well each of these properties are decoded from each model aligns with patterns in how well each of these properties are decoded from specific brain regions.
>     - In particular, the ranked performance of the MD system in decoding these properties is matched perfectly by several complex transformer-based models, whereas the ranked performance of the LS in decoding these properties is matched perfectly by simpler, token-based models like `bag-of-words` and `TokenProjection`.
>
> Taken together, our results do seem to suggest that the MD system is distinctly involved in encoding properties like dynamic analysis and those properties captured by complex model architectures like transformers, while the LS is involved in encoding token-level information. This is a new and concrete result which our experiments offer.
>
> That said, our approach will not be able to determine whether the mechanisms employed in our brains to process and understand code mimic the learning objectives of code models. We can perhaps only find initial evidence for such an effect, which will then have to be confirmed using more systematic analyses and causal manipulations.
>
> We have highlighted this discussion in section 6 in the modified draft (marked in blue). We will highlight this specific question and enhance our discussion in the camera-ready draft, where we will have access to one more page.
>
> ----------------

---

### Official Review · Reviewer_jQhV · 2022-07-12

**Rating:** 3
**Confidence:** 4
**Soundness:** 4 excellent
**Presentation:** 3 good
**Contribution:** 1 poor

**Summary:**

In this paper the authors show that they can decode various properties of computer code (e.g. python programs) from brain images taken while subjects read and try to interpret those programs. More specifically, the authors separate the brain data into language and multiple-demand (MD) networks and show that (1) many code properties are represented in both networks, but (2) more information about several properties is represented in the MD network.


**Questions:**

I would challenge the authors to go a little further afield with their analyses, which currently hew very closely to standard practices. Can anything be said about what kinds of properties the different self-supervised models represent? Do they capture similar information to the theoretically-defined properties? Is there any relationship between different properties captured by the models and representations in different brain areas? This seems like a rich and underexplored area, so it is a bit frustrating to see this paper do such shallow analyses!


**Limitations:**

None noted.

**Strengths And Weaknesses:**

Strengths:
* Overall the paper is fairly clear, the analyses are well-motivated, and the results are well-explained.
* The methods are clearly and thoroughly explained (bravo!).
* The statistical analyses are of high quality and well-motivated.

Weaknesses:
* Methodologically, this paper is very simple. Oftentimes the best approach is the simplest (linear decoders DO work better than most other approaches on fMRI data), especially when it yields a compelling result. But here the results are, unfortunately, not very compelling, and I struggle to find their significance. There are some small differences between the findings here and those from earlier work on this dataset (Ivanova et al., 2020), particularly in assigning a more general role during code comprehension to the language network. However, it’s not clear how this should be interpreted in light of other knowledge about the language network and its representations.
* The most methodologically interesting analysis, in which code representations from self-supervised sequence models are compared to the brain, is underdeveloped. It seems like something much more interesting could be said about what these models represent, and how those representations relate to the MD and language networks.

---

> ### Author Response · Authors · 2022-08-01
> **Response to Reviewer jQhV - part 2**
>
> > 1. Can anything be said about what kinds of properties the different self-supervised models represent? Do they capture similar information to the theoretically-defined properties?
>
> Great question!
> This analysis is available in Table 4 , Appendix C.2 of the submitted draft
>
> To evaluate this, we probed each code model for the same set of properties as the brain regions. We observed outstanding success, with classification and regression results often approaching perfect correlation.
>
> We conclude that the properties we explore, among others, are encoded by these models.
>
> ----------------
> > 2. Is there any relationship between different properties captured by the models and representations in different brain areas?
>
> While each of the properties evaluated in this manuscript are strongly decoded from each of the code models explored, there is some fine-grained variability with respect to which properties are best decoded from which models. In order to assess whether this fine-grained variability maps onto the content of specific brain regions, we perform a new analysis.
>
> For each brain region and code model, we extract the $z$-score of how well each hand-selected property is decoded. We then compare the performances of each brain region to each code model via a Spearman rank correlation (details in the newly added Appendix G of the revised draft). Under such a framework, if two representations have similar content, and perform similarly in how well or how poorly they decode a battery of properties, they will show strong positive rank correlations. On the other hand, if two representations show different patterns in how well they perform on the same tasks, they will be less strongly correlated.
>
> Here, we find that 3 transformer architectures: CodeBERTa, CodeBERT, and CodeTransformer, show perfect decoding performance rank correlation with the MD system. On the other hand, bag-of-words and TokenProjection, which only reflect the presence of specific tokens, show perfect rank correlation with the Language System.
>
> These data highlight two distinct computational motifs among computational models of code: those that track tokens and develop LS-like representation content, and those that represent higher-level MD-like content, possibly through contextualization and interaction between program components.
>
> **This is a new finding, which strongly supports our earlier reported results of the MD system being distinctly involved in encoding properties like dynamic analysis and those properties captured by complex model architectures like transformers.**
>
> ----------------

---

> > ### Comment · Reviewer_jQhV · 2022-08-08
> > **Response to rebuttal**
> >
> > Thank you for the thorough and clear responses to my review. The new analysis (Appendix G) and relevant existing analysis (Appendix C.2) do address the questions posed in my original review. But the results from these analyses, unfortunately, do little to clarify the relationships between the different representations that are being tested.
> >
> > > To evaluate this, we probed each code model for the same set of properties as the brain regions. We observed outstanding success, with classification and regression results often approaching perfect correlation.
> > > We conclude that the properties we explore, among others, are encoded by these models.
> >
> > I had hoped that the model probing would help interpret the model decoding results, as some properties might be better captured by some models. But the conclusion here is that all models capture all properties. This result is a little difficult to square with the main analyses in the paper. The MD system can decode dynamic analysis properties, and can decode the transformer-based models (CodeBERTa, CodeTransformer) better than other models (Token Projection), but all of the models capture all of the code properties? The Token Projection model even captures the dynamic analysis properties better than the transformer-based models! This set of results doesn’t clarify what we can infer from the model decoding results.
> >
> > > Here, we find that 3 transformer architectures: CodeBERTa, CodeBERT, and CodeTransformer, show perfect decoding performance rank correlation with the MD system. On the other hand, bag-of-words and TokenProjection, which only reflect the presence of specific tokens, show perfect rank correlation with the Language System.
> >
> > This result is also difficult to square with the main decoding analyses, which showed that MD is better at decoding every representation (although only significantly better for some) than LS. Is this new result proposing that the simple token-level models are a better match to LS in some way? If so, why is MD better able to decode to those same models? Further complicating the picture, the TF-IDF model, which is also token-level, is better correlated with the MD decoding than LS (and is significantly better decoded by MD than LS). Why is this model a better match for MD than the other token-level models (Token Projection, Bag Of Words)?
> >
> > I am not arguing here that any of these results are incorrect, but they do present a confusing mish-mash. This suggests to me that something about the models or methods here is simply not “carving nature at its joints”. The models and properties _do_ seem to capture something about how code is represented in the brain, and there _do_ seem to be differences between brain areas as well as differences between models/properties in terms of how they are represented. But without further insight into what the models are doing or how specific code properties are represented in the brain, this story is incomplete.

---

> > > ### Author Response · Authors · 2022-08-09
> > > **Thanks**
> > >
> > > Thank you for your reply.
> > >
> > > We are glad that you found the new analyses do address the questions posed in the original review.
> > >
> > > We too had hoped that the model probing results would have provided more context to the core brain analyses, but indeed all code properties are represented by each code model.
> > > While the decoding of all properties by all code models is high, we do find some context in the relative performance of each model at decoding each property, as we elaborate in Appendix G.
> > >
> > > > Is this new result proposing that the simple token-level models are a better match to LS in some way?
> > >
> > > What this new result reveals is a comparison in the decodable content from the representations, through the set of code properties we explore.
> > > Both the LS and token-level models show the same ranked ordering in how well they represent each of the code properties.
> > >
> > > The discussion on TF-IDF is indeed complex.
> > > While it does represent tokens, it also contextualizes for corpus-level distributional information, and as such does not purely reflect only the presence of specific tokens, as in the other case.
> > > We agree this distinction is subtle, find this result suprising, and we are working to comprehensively understand this model space and address these fine-grained comparisons.
> > >
> > > > But without further insight into what the models are doing or how specific code properties are represented in the brain, this story is incomplete.
> > >
> > > We would love to hear any specific feedback you may have on what experiments may shed more light on this relationship in the context of our current setup.
> > >
> > > In all, we present the first set of results reflecting a correspondence between the representations of brain regions and code models, including a subset of relevant properties in explaining that comparison. While not every data point of analysis lends itself to a neat interpretation, we do believe that this set of results provides valuable initial insights worth building upon by a larger community, and we release a flexible and comprehensive codebase for new users to test different code properties and models.
> > >
> > > We hope that you find the contribution of this work worth consideration, and we thank you for your excellent and thought-provoking questions in this discussion, which have improved the current manuscript and will guide targeted follow-up work.

---

> ### Author Response · Authors · 2022-08-01
> **Response to Reviewer jQhV - part 1**
>
> Thank you for this review, and we are glad that you found our method thoroughly explained.
>
> We were inspired by your question on understanding the effect of code model representation and code properties and carried out an additional analysis which sheds new light (details below, after the section _summary_). That said, we believe we have analyzed the data sufficiently to learn the most from it while not overclaiming. We have many additional analyses documented in the appendix which we did not get a chance to discuss in the main draft owing to space constraints; we restrict our discussion to analyses which did yield significant insight.
>
> Concretely, we analyzed the following:
>
> - ANOVA to assess whether decoding performance varies significantly across brain regions for each code property and code model (appendix D).
> - ANOVA to assess whether decoding performance varies significantly across code models for each brain region (appendix D)
> - Pairwise comparisons between brain regions for code property and code model decoding results and between code properties and code models for their decodability from brain regions. (appendix E)
>
> These three experiments form a part of the main results we report in Sections 5.1, 5.2. We investigated to see if there exists reliable patterns in which code properties brain regions most sensitively and selectively decode, and to which code models they most effectively map.
>
> In addition to this set of tests, we also investigated:
> - Representation of code properties in different brain regions (appendix C): a question that you also raise.
> - The relative content of each brain region representation above and beyond other regions using a partial regression framework (appendix F).
> - Inter-correlations between code properties over programs to assess their distinctness in our dataset (appendix H).
> 	- The properties we evaluate are not strongly confounding.
> - Sensitivity of the dimensionality of code model embeddings as decoding targets (appendix I).
> 	- lower complexity models appear to favor lower dimensional embeddings, higher complexity models suffer more from compression.
> - Sensitivity to metric choice used in the code property decoding results: Pearson correl vs. RMSE (appendix J).
>
> We summarize below our key results from our submission, and add to it a new set of observations which another analysis yielded.
> We have made brief updates to sections 5 and 6 of the submitted draft which discuss these observations (marked in blue font).
> We have also added Appendix G which provides details of this additional analysis (and will pull more of this information into the main text under the assumption that the camera ready version will allow us an additional page)
>
> We hope that you find this addition informative, and consequently of higher contribution and impact.
>
> We are happy to engage further to improve the quality of this discussion.
>
> ----------------
>
> **A summary of our results**
>
>
> - _Brain regions represent code properties._ (Section 5.1 in submitted draft)
>
>     - The MD and LS encode properties about comprehended code, including `control flow`, `datatype`, `static analysis`, and `dynamic analysis`.
>     - The MD decodes `dynamic analysis` properties $>$ `static analysis`
>     - The `dynamic analysis` property is decoded from MD $>$ LS
>
> - _Brain region representations linearly map to code model representations._ (Section 5.2 in submitted draft)
>
>     - The MD and LS linearly map to a collection of code models, spanning from a simple `TokenProjection` to several transformer architectures.
>     - The MD maps to the representations of 3 models more effectively than to the `TokenProjection` model, lending evidence that the MD encodes information beyond the presence of specific tokens. We do not find such evidence for the LS.
>
> - _Code models and brain regions show correspondence in the properties that can be most-effectively decoded from them._ **(new analysis; added to section 5.2 and Appendix G in the revised draft)**
>
>     - The presented code models encode all the code properties we investigate.
>     - Variation in how well each of these properties are decoded from each model aligns with patterns in how well each of these properties are decoded from specific brain regions.
>     - In particular, the ranked performance of the MD system in decoding these properties is matched perfectly by several complex transformer-based models, whereas the ranked performance of the LS in decoding these properties is matched perfectly by simpler, token-based models like `bag-of-words` and `TokenProjection`.
>
> ----------------
>
> We respond to specific questions below.

---

### Official Review · Reviewer_Mhum · 2022-07-12

**Rating:** 7
**Confidence:** 5
**Soundness:** 3 good
**Presentation:** 3 good
**Contribution:** 3 good

**Summary:**

This paper studies code comprehension in the human cortex by evaluating the ability of four different functional networks (multiple-demand, language, auditory, visual) to decode different code properties. The main questions posed are:
- What is the functional selectivity of different brain regions? (for ex., syntactic or semantic code properties)
- Do different regions encode similar properties?

The main results include:
- Differential selectivity for code properties between the language and multi-demand systems.
- Dependence of decoding accuracy on token-level information
- No significant role of the auditory cortex as expected and the processing of low-level properties by the visual cortex.



**Questions:**

1. Could the authors elaborate on the significance testing procedure and what “labels” refers to in line 213? Relatedly, what is the significance test used to identify significant  significant effects in Analysis 2 (multi-system partial regression analysis)? Although not identical, this is similar to variance partitioning done in encoding models and the paper could benefit from adding a citation to Deniz et al., 2019.
2. The difference in properties encoded better by MD vs. LS is very interesting! (lines 329-338) What do the authors make of these differences in terms of hypothetical roles of each system in processing code? Can this be related, perhaps, to functional specificity found in MD & LS for language or the timescales of processing? The paper would benefit from some discussion on the same.
3. The results presented in fig 3, panel 2 are also interesting! In my opinion, it is not apparent that the auditory cortex is performing above baseline solely because of token-level matches + refer to comments on clarity in previous section.
4. Is this following necessarily true? “It is currently unclear what mechanisms drive code comprehension. If we find one class of ML models (say, masked language models) to be more predictive than another (say, autoencoders), **it is reasonable to suspect that our brains optimize objectives more similar to that of masked-LMs than that of autoencoders when comprehending code.** ” I would argue that the brain and machines need not be optimizing the same objective to capture similar information. Without systematic analyses of replicated behaviors, prediction performance is not solely enough to answer such a question.
5. Was the right hemisphere left out from the language system? If so, why? (there has been a huge body of work in the past decade showing evidence of robust, reliable RH activity during language comprehension)


**Limitations:**

-NA-

**Strengths And Weaknesses:**

Edit after rebuttal: I thank the reviewers for their response. I believe they have sufficiently answered my questions and recommend this paper for acceptance.

##################################################################################################

Strengths:

1. Clarity (of writing): the paper was very well written and easy to understand.
2. Originality & Significance: Functional selectivity in code comprehension is under explored and this paper effectively investigates this across multiple systems and properties.
3. Quality: The analyses presented here are thorough and analyze several possible sources of confounds/relationships between the systems and feature spaces. I believe that the claims made are well supported by the evidence.
4. Misc.: All the data and code will be made freely available.

Weaknesses:
1. Clarity (of visualizations & results): Since the paper conducts several analyses and uses multiple baselines, the methodological details were lost on me at several places. For example, what is the significance test being in used in panel 2 of FIg. 3 and how is this different from the token projection baseline in the interpretation? (since the paper reports significant performance over chance even in cases where the performance is no different from the token projection features). Furthermore, the authors make several key, interesting observations in the text but this is not apparent from looking at Fig. 3 alone. For example, the preference of static vs. dynamic features between MD & LS or the models that performed significantly better than token projection. I understand the severe space constraints, but I would urge the authors to try alternate visualizations for the main paper that highlight the results presented in the text.

NB: I have reviewed a previous version of this paper and am happy to see that the authors have considerably revised the analyses and manuscript.

---

> ### Author Response · Authors · 2022-08-01
> **Response to Reviewer Mhum - part 2**
>
>
> > 5. The difference in properties encoded better by MD vs. LS is very interesting! (lines 329-338) What do the authors make of these differences in terms of hypothetical roles of each system in processing code?
>
> We are glad that you found this interesting, and we agree!
>
> Indeed, differences in the functional specificity for language relative to domain-general processing definitely play a role here, and we briefly discuss the notion of the language network as parsing tokens while the MD network performs simulation (Section 6).
>
> An additional supplemental analysis in Appendix G adds to the evidence supporting this idea, where we observe the fine-grained decoding performance from the LS representations is consistent with that of token-based models, whereas the MD performance is more consistent with that of transformer-based architectures. We discuss this in sections 5 and 6, which we have modified as part of the rebuttal (marked in blue).
>
> ----------------
>
>
> > 6. The results presented in fig 3, panel 2 are also interesting! In my opinion, it is not apparent that the auditory cortex is performing above baseline solely because of token-level matches
>
> Indeed, the auditory cortex does not perform above the null baseline for any of the properties presented in Figure 3 panel 2 (as noted by the lack of asterisks at the base of each bar in our revised Figure 3).
>
> ----------------
>
> > 7. I would argue that the brain and machines need not be optimizing the same objective to capture similar information. Without systematic analyses of replicated behaviors, prediction performance is not solely enough to answer such a question.
>
> We agree.
>
> However, a significantly stronger predictive performance favoring one objective over another is definitely indicative of some preference (irrespective of what the specific objective is which the brain actually encodes).
>
> We have only just begun finding evidence for such an effect in language (Caucheteux & King, 2022).
> It would definitely be an interesting result to explore further, through more systematic analyses and causal manipulations.
>
> ----------------
>
> > 8. Was the right hemisphere left out from the language system? If so, why?
>
> We agree that RH language activity is worth exploring when studying language.
> In our specific case though, we study only the LH language system because we did not find any significant activity in the RH on doing a global searchlight analysis for the Code > Sentences contrast, which was also confirmed both by Liu et al. (2020) and Ivanova et al. (2020).
>
> ----------------

---

> ### Author Response · Authors · 2022-08-01
> **Response to Reviewer Mhum - part 1**
>
> Thank you for your further review of our paper. We appreciate that you have found this version to be original, significant, and of high-quality. We summarize the changes we’ve made below, and follow that up with addressing specific questions you raise.
>
> **Summary of changes made**
>
> - Revised Figure 3, which now clarifies the auditory cortex’s performance, and the significance of results wrt token projection baseline.
> - Additional supplementary analysis (added in Appendix G) which discusses the difference in properties encoded better by MD vs. LS.
> - We discuss these newer results in sections 5, 6 in our modified draft. The changes have been marked in blue.
>
>
> **Answers to your questions**
>
> > 1. The authors make several key, interesting observations in the text but this is not apparent from looking at Fig. 3 alone.
> Thank you for your assessment of Figure 3, and the opportunity to improve the clarity of our presentation. Due to space limitations, we have only included a subset of the significance tests from our paper in this core multi-panel figure, so as to highlight the most critical pieces of information.
>
> The line over panel 2 marking out `MD Dynamic Analysis` from `LS Dynamic Analysis` and `MD Static Analysis` denotes the significance of those paired two-sample $t$-tests ($p$<0.05; FDR-corrected) as described on lines 307-313 in the submitted draft.
>
> The statistical difference of each code model from the `TokenProjection` baseline was not reflected in the core figure, which we have now updated.
>
> Additionally, we will enlarge and break this figure up, and add more clarifying details in the additional page made available in the camera-ready version.
>
> ----------------
>
> > 2. Could the authors elaborate on the significance testing procedure and what “labels” refers to in line 213?
>
> On line 213, “labels” refer to the decoding tasks, e.g. the class label of `math` or `str` in the case of the `datatype` property.
>
> By keeping `X` (the neural recordings) the same, and shuffling the rows of `Y` (the code properties or models), we establish a non-parametric null distribution for each individual decoding task (from each brain network to either each code property or code model representation).
> We then fit a univariate Gaussian distribution to the sampled null, using the sample mean and variance, and evaluate the tail-probability of the non-permuted decoding score so as to assess whether or not it differs meaningfully from this null-baseline.
>
> A detailed description can be found in Appendix B - Method Details, lines 758-766 in the submitted draft.
>
> ----------------
>
>
>
> > 3. Relatedly, what is the significance test used to identify significant effects in Analysis 2 (multi-system partial regression analysis)?
>
> In _Analysis 2, partial regression analysis_ (section 5.1), we use the same paired two-tailed $t$-test ($p<0.05$; FDR-corrected) employed in the rest of the manuscript.
>
> Using this test, we compare the decoding score of a given property when both `MD` and `LS` were included as features in the linear model relative to when only one of the two was included.
>
> ----------------
>
> > 4. Although not identical, this is similar to variance partitioning done in encoding models and the paper could benefit from adding a citation to Deniz et al., 2019.
>
> Yes, this bears a resemblance to variance partitioning done in encoding models as both stem from the same statistical framework of incremental partial regressions.
> We have added a citation to Deniz et al., 2019. Thanks!
>
> ----------------

---

### Meta-Review · Area_Chair_JPSY · 2022-08-26

**Recommendation:** Accept
**Confidence:** Less certain

**Metareview:**

This work examines the relationship between fMRI recordings of people who read short programs and different properties and representations of the programming code.
The aim of the work is to understand what properties of code are encoded by different brain systems, and to understand how similar the representations of code in the brain are to those encoded by self-supervised language models that are pretrained to encode programming code.
More specifically, the authors separate the brain data into language (LS) and multiple-demand (MD) networks and show that (1) many code properties are represented in both networks, but (2) more information about several properties is represented in the MD network.

The paper presents a fairly original idea, namely that of generalizaing brain encoding of language to code.
The work presents the thorough experimentation and improved discussion and motivation: thorough discussion of the literature, good motivation of the analyses, sound interpretation of the results, with clear and concise writing.
What is less clear overall is the results, that may not bring as much insights as one would hoped. There are some small differences between the findings here and those from earlier work on this dataset (Ivanova et al., 2020), particularly in assigning a more general role during code comprehension to the language network. Yet these differences are hard to interpret in view of current knowledge.
Furthermore, the MD system is overall better at decoding every representation than LS, which does not reveal specific functional characteristics. On the other hand, novel experiments show that simple token-level models are a better match to LS, while the TF-IDF model, which is also token-level, is better correlated with the MD decoding than LS. These results hardly provide a consistent picture.

Besides, one may wonder whether the strategy of focussing on such networks (that are not as homogeneous as claimed by the authors) is really a good one.

For this reason, there is a large variance between reviewers ---and a very long discussion, which remained open. In particular, the authors are providing new results in Appendix, which complement the ones described in the original submission, but are also hard to gather with the original results.

Overall, my feeling is tha the paper opens an interesting direction, and could be accepted at NeurIPS to further feed the discussion.


**Award:**

No

---

### Decision · Program_Chairs · 2022-09-14

Accept